# High-resolution deep mutational scanning of the melanocortin-4 receptor enables target characterization for drug discovery

Conor J Howard[1†], Nathan S Abell[1†], Beatriz A Osuna[1§], Eric M Jones[1], Leon Y Chan[1], Henry Chan[1], Dean R Artis[1#], Jonathan B Asfaha[1], Joshua S Bloom[2,3], Aaron R Cooper[1¶], Andrew Liao[1], Eden Mahdavi[1], Nabil Mohammed[1], Alan L Su[1**], Giselle A Uribe[1], Sriram Kosuri[1], Diane E Dickel[1*‡], Nathan B Lubock[1*‡]

[1]Octant, Inc, Emeryville, United States; [2]Department of Human Genetics and Department of Computational Medicine, University of California, Los Angeles, Los Angeles, United States; [3]Howard Hughes Medical Institute, Chevy Chase, United States

*For correspondence:
diane@octant.bio (DED);
nate@octant.bio (NBL)

†These authors contributed equally to this work
‡These authors also contributed equally to this work

Present address: §Tacit Therapeutics, Inc, South San Francisco, United States; #Annexon Biosciences, Brisbane, United States; ¶Arsenal Biosciences, South San Francisco, United States; **Department of Chemical and Systems Biology, Stanford University, Stanford, United States

## eLife Assessment

The authors use deep mutational scanning to assess the effect of ~6,600 protein-coding variants in MC4R, a G-protein-coupled receptor associated with obesity. They develop new, more precise approaches to deep mutational scanning, enabling them to probe molecular phenotypes directly relevant to the development of drugs that target this receptor. In this **important** work, the authors provide **compelling** evidence that variants impact signaling through MC4R in different ways, that some defective variants are amenable to a corrector drug and that deep mutational scanning data could guide compound optimization.

**Abstract** Deep Mutational Scanning (DMS) is an emerging method to systematically test the functional consequences of thousands of sequence changes to a protein target in a single experiment. Because of its utility in interpreting both human variant effects and protein structure-function relationships, it holds substantial promise to improve drug discovery and clinical development. However, applications in this domain require improved experimental and analytical methods. To address this need, we report novel DMS methods to precisely and quantitatively interrogate disease-relevant mechanisms, protein-ligand interactions, and assess predicted response to drug treatment. Using these methods, we performed a DMS of the melanocortin-4 receptor (MC4R), a G-protein-coupled receptor (GPCR) implicated in obesity and an active target of drug development efforts. We assessed the effects of >6600 single amino acid substitutions on MC4R's function across 18 distinct experimental conditions, resulting in >20 million unique measurements. From this, we identified variants that have unique effects on MC4R-mediated $G\alpha_s$- and $G\alpha_q$-signaling pathways, which could be used to design drugs that selectively bias MC4R's activity. We also identified pathogenic variants that are likely amenable to a corrector therapy. Finally, we functionally characterized structural relationships that distinguish the binding of peptide versus small molecule ligands, which could guide compound optimization. Collectively, these results demonstrate that DMS is a powerful method to empower drug discovery and development.

## Introduction

Deep Mutational Scanning (DMS) employs cutting-edge synthetic biology or genome editing methods, DNA synthesis, and sequencing to systematically assess the effect of every possible single amino acid substitution on the function of a protein target (*Araya and Fowler, 2011*; *Fowler and Fields, 2014*; *Starita et al., 2017*). Researchers have leveraged DMS to gauge a variety of protein functions or their cellular consequences, including viability (*Findlay et al., 2014*), protein abundance (*Faure et al., 2022*; *Matreyek et al., 2018*), transcriptional signaling (*Jones et al., 2020*), and inter- and intra-molecular interactions (*Braberg et al., 2022*; *Faure et al., 2022*). DMS assays are increasingly used for human variant interpretation (*Weile and Roth, 2018*) and to elucidate the relationship between protein structure and function, including in the evaluation (*Brandes et al., 2023*) and finetuning of protein language models (*Lafita et al., 2024*).

While DMS has significantly advanced our understanding of protein function, its potential in drug discovery and development has yet to be fully realized. Applications in this realm require methods that are more disease-relevant, sensitive, and quantitative to measure the subtle effects of both sequence variants and experimental conditions (e.g. drug treatments). For example, DMS assays often measure effects like viability, which are several biological layers removed from the specific molecular mechanisms modulated by drugs. Additionally, the current signal-to-noise ratios of many assays and challenges around uncertainty quantification make drawing quantitative conclusions from DMS data difficult. Moving from categorical (e.g. benign vs. pathogenic) classifications toward precise quantitative measurements of human variants could improve predictions of safety and efficacy at early stages of drug development programs (*Plenge et al., 2013*). It would also expand the use of matching patients' medications to their specific genetic variants (i.e. theratyping), which could lead to better patient outcomes (*McDonald et al., 2024*). Finally, DMS data contain the functional consequences of thousands of different biochemical perturbations. Sufficiently sensitive assays against functional readouts and improved analysis methods would readily complement structure-based approaches by elucidating the functional consequences of ligand binding. Building on this, it should be possible to identify novel protein-ligand interactions that could be reverse-engineered to increase compound potency, further expanding the potential impact of DMS on drug discovery.

In a previous work, we described DMS methods to measure the effects of thousands of single amino acid variants on the function of the beta-2 adrenergic receptor (β2AR; *Jones et al., 2020*), a member of the G-protein-coupled receptor (GPCR) class that is the most commonly targeted protein family in drug development (*Sriram and Insel, 2018*). Here, we build upon this previous work to demonstrate more sensitive and robust DMS methods for drug discovery and development, focusing on the melanocortin-4 receptor (MC4R). MC4R is a GPCR, and human variants that result in partial or complete loss of its function cause the most common form of inherited obesity [OMIM #618406] (*Farooqi et al., 2003*; *Hinney et al., 2003*; *Vaisse et al., 2000*). Variants that increase MC4R activity are protective against obesity (*Lotta et al., 2019*; *Paisdzior et al., 2020*), and numerous small molecule and peptide agonists of MC4R have been tested as potential therapeutics (*Chen et al., 2015*; *Clément and van den Akker, 2020*; *Collet et al., 2017*; *Greenfield et al., 2009*; *Hinney et al., 2022*; *Huang and Tao, 2014*; *Kievit et al., 2013*; *Sweeney et al., 2023*).

In this study, we developed substantially improved experimental and analytical methods for DMS that are capable of detecting subtle quantitative effects of variants and differences between experimental conditions with a high degree of statistical rigor. We then tested the effects of nearly all possible single amino acid variants of MC4R (6633 of 6640) on two distinct GPCR signaling functions under a variety of treatment conditions. From this, we generated a high-resolution map of how MC4R's structure relates to function, and we accurately classified the quantitative effects of human variants. Additionally, we identified amino acid changes that differentially impact (i.e. bias) MC4R's different GPCR signaling functions, pinpointed human variants that are amenable to a specific class of therapy, and elucidated the functional impact of protein-ligand interactions between MC4R and both peptide and small molecule ligands. This demonstrates the utility of DMS for various drug discovery and development applications, and the methods described herein should be broadly applicable to GPCRs and other drug target classes that function in transcriptional signaling pathways.

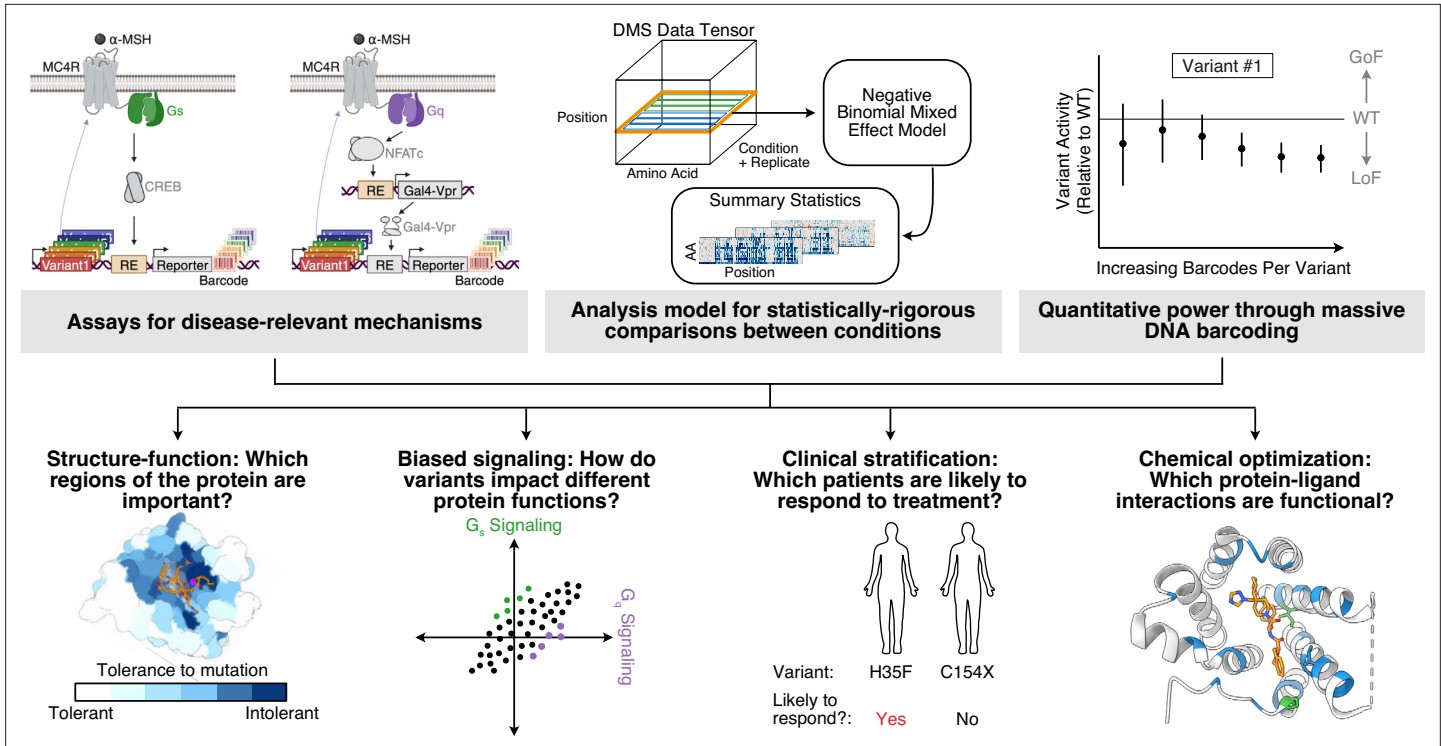

**Figure 1.** Highly quantitative Deep Mutational Scanning (DMS) methods for drug discovery applications. Top: DMS experimental and analysis methods improvements reported in this paper. Bottom: Questions commonly encountered in drug discovery and development that can be addressed by improved DMS methodology. Panel A created with BioRender.com.

The online version of this article includes the following figure supplement(s) for figure 1:

**Figure supplement 1.** Methods development detail.

**Figure supplement 2.** Analysis method and barcoding detail.

## Results

### Development of highly quantitative deep mutational scanning methods

#### Assays for disease-relevant mechanisms

A critical tool in drug discovery programs is a very sensitive set of assays that directly interrogate the function(s) of a protein target against which to test potential therapies (*Hughes et al., 2011*). This is essential for ensuring that compounds are modulating a specific mechanism underlying disease and not having undesired 'off-target' effects. Stimulation of MC4R with its agonist, alpha melanocyte-stimulating hormone (α-MSH), results in signaling through multiple canonical GPCR pathways, including Gα$_s$-coupled cyclic adenosine monophosphate signaling (hereafter referred to as Gs) and Gα$_q$-coupled calcium signaling (hereafter referred to as Gq; *Tao, 2010*). Therefore, we first developed multiplexed reporter assays for these two critical MC4R G-protein signaling functions (*Figure 1*, *Figure 1—figure supplement 1*), building off of our earlier work performing a deep mutational scan of β2AR (*Jones et al., 2020*). Both reporters were designed for use in human HEK293T cells and to be compatible with our previously described DMS library construction methods (*Jones et al., 2020*). Briefly, these methods harness high-throughput DNA synthesis to construct every possible single amino acid variant, and each variant is then linked to a transcriptional reporter containing an oligonucleotide sequence barcode unique to that variant. Reporter constructs are then integrated into cells using a site-specific recombination-based landing pad system and drug selection to ensure that each cell contains a single variant-barcode combination. Activation of the receptor turns on a response element for the signaling pathway, leading to the expression of the barcoded reporter, which is then quantified using RNA sequencing.

The MC4R Gs assay was adapted and further optimized from the cAMP response element-based reporter we previously used for β2AR (*Figure 1—figure supplement 1A and C*). For Gq signaling,

an analogous reporter using an NFAT response element (*Boss et al., 1996*) alone to activate reporter gene expression was not suitable due to weak signal-to-noise (*Figure 1—figure supplement 1D*). To solve this problem, we incorporated a 'relay' system to amplify the reporter signal using a synthetic transcription factor composed of Gal4 fused to the VP64-p65-Rta (VPR) transcriptional activator (*Chavez et al., 2015*; *Figure 1*, *Figure 1—figure supplement 1B*, see Materials and methods for more details). The resulting Gal4-VPR transcription factor was placed under the control of the NFAT response element, and the reporter gene was placed under the control of a UAS element that responds to Gal4 binding. This assay design resulted in robust reporter expression upon stimulation of MC4R (*Figure 1—figure supplement 1C*). Together, these assays provide sensitive functional readouts for two important MC4R signaling activities implicated in obesity and other phenotypes (*Fatima et al., 2022*; *Sweeney et al., 2023*), and we used them to build DMS libraries to assess the effects of all possible single amino acid substitutions in MC4R.

## Analysis model for statistically robust comparisons

To explore many hypotheses that arise in drug discovery applications, it is valuable to assay a DMS library using experimental replication and under a variety of conditions, such as different drugs and/ or pathways of interest. However, most popular methods for DMS analysis do not leverage DNA barcodes or other experimental replicate information, nor do they support hypothesis testing between conditions (*Faure et al., 2020*; *Rubin et al., 2017*). Consequently, we developed an alternative modeling framework to enable this (*Figure 1*, *Figure 1—figure supplement 2A*, see Materials and methods: Negative binomial regression analysis pipeline for additional details). Briefly, borrowing from approaches for inferring differential expression from RNA-seq data (*Ahlmann-Eltze and Huber, 2021*; *Love et al., 2014*; *McCarthy et al., 2012*), we applied a mixed effect negative binomial generalized linear model (GLM) to raw barcode counts directly. The model contains a random effect across barcodes to share barcode information between replicates and conditions, and incorporates sample-specific offsets to account for technical covariates like sequencing depth, as is common for RNA-seq (*Robinson and Oshlack, 2010*). For each variant, we estimate the mean shift in barcode count and associated standard error for each treatment condition, relative to wild-type. Using per-condition summary statistics, we either directly test whether each variant barcode mean is significantly different from wild-type (zero) in each treatment, or we can define more complex linear contrasts on variant effects across multiple treatments.

## Increasing power through barcoding

As described above, our assay design and analysis framework harness DNA barcodes that are uniquely associated with a particular variant and provide multiple independent measures of a variant's effect. In our previous DMS of $\beta_2$AR, the median number of barcodes independently linked to each variant was ~10 (*Jones et al., 2020*), and we reasoned that increasing this number would increase the power to detect functional effects. To this end, we optimized and scaled our library cloning and cellular integration protocols to target ~30 barcodes per variant in building DMS libraries for MC4R. As expected, this increased the power to detect variant effects. For example, the separation between the activity of alleles that are clearly deleterious (i.e. a stop codon at any position in the protein) and all other alleles (i.e. wild-type or missense variants) was drastically increased in the MC4R Gs assay relative to the same assay for $\beta_2$AR (*Figure 1—figure supplement 2B*). To further test the effect of the number of barcodes for a given variant, we computationally down-sampled the barcodes for representative positions of MC4R and ran the resulting data through our analysis pipeline. As expected, by increasing the number of barcodes per variant, the magnitude of the standard error of the estimated variant effect decreases in a manner consistent with increased sample size (*Figure 1—figure supplement 2C*). This confirms that increasing the number of barcodes per variant enables the quantification of subtle differences within a standard hypothesis testing framework, and provides an experimental parameter that one can vary to improve the power to detect the effects of sequence variants of particular interest.

## Comprehensive deep mutational scanning of MC4R

With these methods in hand, we carried out a comprehensive assessment of the effects of all single amino acid substitutions (including nonsense variants) on MC4R's Gs and Gq signaling activities under a variety of experimental conditions (*Figure 2a*, *Supplementary file 1a*, *Figure 2—figure supplements*

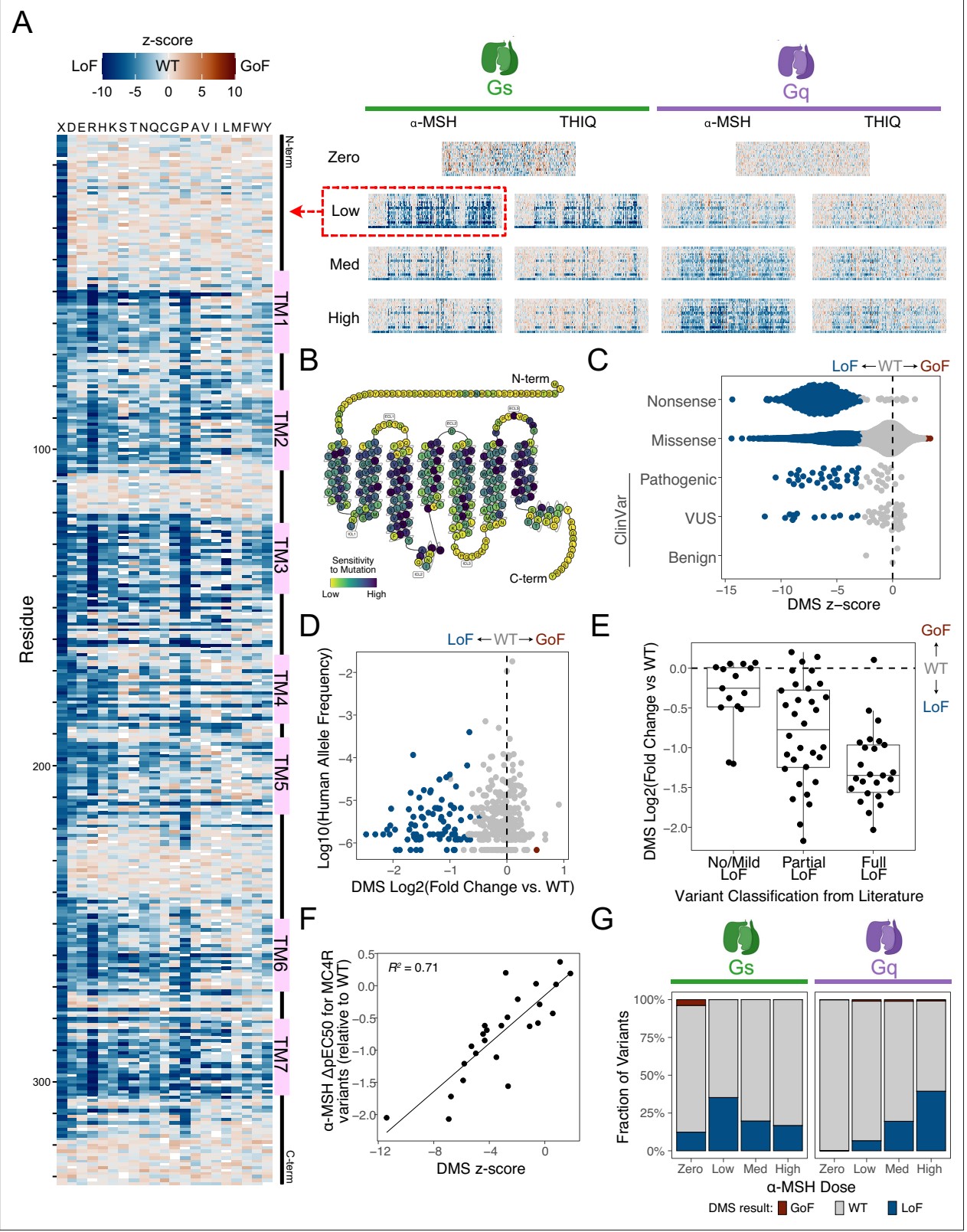

**Figure 2.** Effect of 6633 missense and nonsense variants on MC4R signaling functions. (**A**) Heatmaps showing the functional effects (z-scores) for nearly all possible amino acid substitutions (6633 of 6640) on MC4R activity for two GPCR signaling functions (Gs and Gq) under a variety of conditions. Heatmaps showing results (both z-score and log₂[fold change of variant activity over wild-type]) for all experimental conditions are shown in *Figure 2—figure supplements 1 and 2*. The results of the Gs assay with low α-MSH stimulation are highlighted on the left (and in Panels B-F). TM: transmembrane

*Figure 2 continued on next page*

*Figure 2 continued*

domain; GoF: gain-of-function; LoF: loss-of-function; WT: wild-type activity (**B**) A modified snake plot (adapted from https://gpcrdb.org/) showing the sensitivity of each MC4R residue to mutation (defined as the mean log$_2$[fold change variant activity over wild-type] divided by sqrt(sum(standard error^2)) after excluding nonsense variants). (**C**) Z-scores for each variant (point), broken out by variant type and clinical (ClinVar) classification. Blue indicates statistically significant LoF, brown is significant GoF (significance threshold: FDR <1%). VUS: variant of uncertain significance. (**D**) Functional effect (log$_2$[fold change of variant activity over wild-type], x-axis) for all human variants relative to the frequency of the allele in the in the human population (y-axis, gnomAD global population, from ***Chen et al., 2024***). (**E**) DMS results for human MC4R variants (y-axis) relative to previous functional classifications in the literature (x-axis, from ***Huang et al., 2017***) (**F**) DMS results (z-scores, x-axis) compared to change in α-MSH potency (relative to WT) of 25 MC4R variants made to the orthosteric binding site for α-MSH (y-axis, cAMP accumulation assay results from ***Zhang et al., 2021***). (**G**) Fractions of MC4R variants that result in LoF, GoF, or WT activity for eight unique experimental conditions.

The online version of this article includes the following figure supplement(s) for figure 2:

**Figure supplement 1.** Results (z-scores) of MC4R Deep Mutational Scans across all agonist stimulation conditions.

**Figure supplement 2.** Results (log$_2$[fold change]) of MC4R Deep Mutational Scans across all agonist stimulation conditions.

**Figure supplement 3.** MC4R Deep Mutational Scans are consistent with variant effect predictions.

---

*1 and 2*). We selected experimental conditions that would inform aspects of drug discovery and development programs, such as elucidating protein structure-function relationships, identifying regions of the protein that bias activity towards or away from a specific function, classifying the effects of human variants in the presence and absence of potential therapies, and uncovering functional differences in protein-ligand interactions. In total, we tested 18 unique conditions, each performed in quadruplicate, including: basal activity (i.e. no stimulation) of MC4R, stimulation of MC4R with a range of doses of the native peptide agonist alpha-melanocyte-stimulating hormone (α-MSH), stimulation with a range of doses of a small molecule agonist (THIQ; ***Sebhat et al., 2002***), treatment with a small molecule corrector (Ipsen 17; ***Poitout et al., 2007***; ***Wang et al., 2014***), and library composition normalization controls (forskolin, see Mterials and methods). Our resulting DMS assays had extraordinary variant coverage, with 99.9% (6,33/6640) of all possible single amino acid substitutions present in all experimental conditions. Each variant was represented by an average of 56 and 28 barcodes for the Gs and Gq signaling pathways, respectively. Between both assays, this translates to more than 557,000 uniquely engineered human cells, each containing a distinctive variant-reporter-barcode combination. When factoring in the number of experimental conditions (18 unique), replicates (four per condition), amino acid variants tested (99.9% of 6640 possible), and the mean barcodes per variant (56 and 28 for Gs and Gq, respectively), this equates to >21,500,000 measurements across all datasets.

Multiple lines of evidence support the high quality and utility of these data for classification of variant effects (***Figure 2A–F***, ***Figure 2—figure supplements 1–3***). Focusing on one dataset as a representative example (Gs signaling using a low dose of α-MSH stimulation), variants that introduce stop codons or fall within transmembrane domains and buried surfaces disproportionately lead to significant loss of MC4R function (***Figure 2A–C***). The results also correlate well with expectations from human genetics data and variant effect prediction algorithms (***Figure 2C and D***, ***Figure 2—figure supplement 3***). For example, the majority (63.3%, 31/49) of human MC4R variants classified as pathogenic or likely pathogenic in ClinVar (***Landrum et al., 2014***) lead to a significant reduction of Gs signaling under low α-MSH stimulation conditions (significance threshold: false discovery rate [FDR]<1%; ***Figure 2C***). Variants that are significantly loss-of-function in this condition are rarer in the human population, and more common human variants have no significant effect on MC4R function (significance threshold: FDR <1%; ***Figure 2D***). Loss-of-function variants by our DMS assay are also typically (e.g. AlphaMissense: 93.4%, 1894/2028) predicted to be deleterious by commonly used variant effect predictors like AlphaMissense (***Cheng et al., 2023***) and popEVE (***Orenbuch et al., 2023***; ***Figure 2—figure supplement 3***).

Because of the sensitivity of our reporter system and the statistical power gained by testing dozens of unique barcodes per variant, we anticipated that these assays would capture subtle quantitative, rather than just qualitative, effects on MC4R function. To assess this, we benchmarked our results against previous quantitative characterizations of MC4R variants from the literature (***Figure 2E and F***). For example, many MC4R variants that have been observed in the human population have been previously tested for their effects on MC4R function, and a review (***Huang et al., 2017***) summarizing this work systematically classified >70 variants according to whether they result in 'full', 'partial', or 'no/mild' loss of MC4R activity. Our results are consistent with these classifications: variants classified

previously as 'full' loss-of-function typically have very low MC4R activity in our assay, 'partial' variants have intermediate effects, and 'no/mild' effect variants have near-normal activity (*Figure 2E*, median log$_2$[fold change of variant activity over wild-type activity] of −1.4, −0.8, and −0.3, respectively, for each group). Finally, our results show a high degree of correlation (Pearson correlation = 0.84, $R^2$=0.71) with quantitative effect measurements reported for 25 variants individually introduced into the orthosteric site of MC4R (*Zhang et al., 2021*). Collectively, this demonstrates that our high quality MC4R DMS data accurately and quantitatively assess the effects of variants on MC4R's function.

### Systematic human variant interpretation of MC4R

Looking across multiple data sets gives a comprehensive picture of variant effects, as the various experimental conditions tested have disparate strengths and weaknesses at detecting loss-of-function (LoF) versus gain-of-function (GoF) activities (*Figure 2G*). For example, unstimulated conditions (e.g. zero α-MSH) uncover variants that lead to constitutive activation of MC4R, but they have less ability to detect loss-of-function variants. In contrast, conditions with agonist stimulation are much better enabled to identify loss of MC4R function. Considering *individual* α-MSH stimulation conditions (zero, low, medium, and high) for both the Gs and Gq assays, each condition identifies 6.6–39.3% of variants as loss-of-function and 0.02–1.1% as gain-of-function (*Figure 2G*). Collectively across all α-MSH stimulation conditions, 3370 variants (50.8%) are loss-of-function in at least one condition, 347 variants (5.2%) are gain-of-function in at least one condition, and 2996 (45.2%) always show wild-type activity. Interestingly, 80 (1.2%) variants are classified as both loss- and gain-of-function, depending upon the condition.

To aid in clinical variant interpretation, we provide detailed functional effect classifications for 220 human variants reported in ClinVar (*Landrum et al., 2014*) or from published patient sequencing studies (*Brouwers et al., 2021*; *Farooqi et al., 2003*; *Hinney et al., 2013*; *Hinney et al., 2006*; *Huang et al., 2017*; *Rodríguez Rondón et al., 2024*; *Stutzmann et al., 2008*; *Wade et al., 2021*; *Yeo et al., 2003*) in *Supplementary file 1b*. In total, 130 of these reported human variants (59.1%) are LoF in at least one condition, consistent with being pathogenic for obesity-related phenotypes. This includes 83.9% (26/31) of the variants that are reported in ClinVar as pathogenic or likely pathogenic, 53.3% (32/60) of those that are unclassified or have conflicting interpretation, and 0% (0/3) of those classified as benign or likely benign. A small number of reported human variants (V103I, H158R, I251L) result in significant increases in Gs and/or Gq signaling, consistent with having a protective effect for obesity, and are generally classified as benign in ClinVar or as having wild-type activity in the literature (*Hinney et al., 2006*). These results highlight the utility of systematic deep scans across multiple experimental conditions for facilitating human variant interpretation.

## Variants that bias MC4R function

MC4R signals through multiple G-protein pathways (*Breit et al., 2011*; *Ju et al., 2018*; *Paisdzior et al., 2020*; *Sharma et al., 2019*), and evidence from human variant interpretation studies (*Lotta et al., 2019*; *Metzger et al., 2024*; *Rodríguez Rondón et al., 2024*), mouse modeling (*Li et al., 2016*), and previous drug programs (*Clément et al., 2018*; *Sharma et al., 2019*) suggest that biasing MC4R's activity toward or away from specific pathways could be therapeutically valuable for the treatment of obesity. To gain a better understanding of how MC4R structure relates to its various functions, we systematically searched for variants that differentially impact Gs versus Gq signaling. We applied Principal Component Analysis (PCA) across eight total DMS datasets (four each for Gs and Gq: with zero, low, medium, and high α-MSH stimulation; see *Supplementary file 1a* for details). The first two principal components explained 66% and 12% of the variance, respectively (*Figure 3A*, *Figure 3— figure supplement 1A and B*). Through inspection, we found that Principal Component 1 (PC1) separates variants that impact both signaling functions, with variants that are loss-of-function for both Gs and Gq having higher PC1 values (*Figure 3—figure supplement 1A and B*). PC2 separates variants that affect Gs and Gq signaling differently (*Figure 3A–D*, *Figure 3—figure supplement 1A and B*). Variants with higher PC2 values typically exhibit Gq bias by having greater than wild-type levels of Gq signaling activity, while retaining wild-type levels of Gs activity (*Figure 3C and D*; *Figure 3—figure supplement 1C and D*). In contrast, variants with more negative PC2 values are Gs-biased variants that typically have wild-type levels of Gs signaling and reduced Gq signaling upon agonist stimulation (*Figure 3C*).

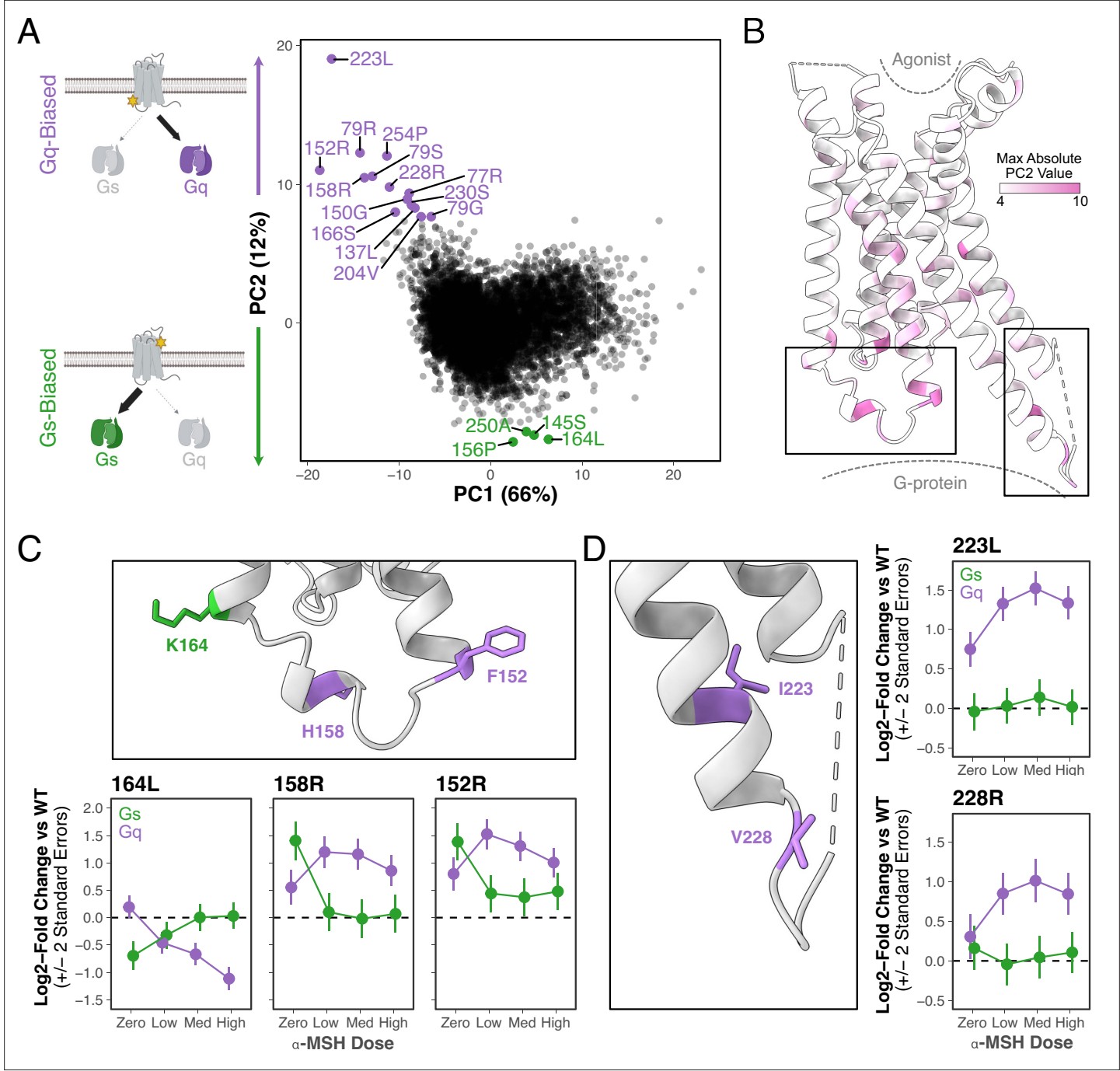

**Figure 3.** Systematic identification of variants that have biased effects on MC4R signaling. (**A**) Principal component analysis of eight MC4R DMS conditions (Gs and Gq signaling each at four α-MSH stimulation doses: zero, low, medium, high, see **Supplementary file 1a**). Each point indicates a unique variant, with those exhibiting extreme Gq or Gs bias colored in purple and green, respectively. PC, principal component. Panel A created using BioRender.com. (**B**) Ribbon structure of MC4R showing the maximum absolute PC2 value for each residue position. (**C**) Top: Closeup of MC4R structure shown in bottom left box in B, highlighting positions where variants result in Gs (green) or Gq (purple) bias. Bottom: MC4R signaling activity (log$_2$[fold change of variant activity relative to wild-type]) for three selected variants across α-MSH doses. Error bars are +/-2 standard errors. Med: medium; WT: wild-type (**D**) Left: Closeup of MC4R structure shown in bottom right box in B, highlighting positions where variants result in Gs or Gq bias as in C. Right: MC4R signaling activity for select variants as in C.

The online version of this article includes the following figure supplement(s) for figure 3:

**Figure supplement 1.** Variants with differential effects on Gs and Gq signaling.

Overall, more variants substantially increase Gq signaling than Gs signaling (*Figure 3A*). Gs is the primary G-protein coupling for MC4R (*Podyma et al., 2018*; *Tao, 2010*), and our data suggest that there is little room for further improving MC4R's already robust Gs signaling activity. Interestingly, biased variants are positionally diverse. For example, the 14 variants that display the most extreme Gq bias (*Figure 3A*) are found at 12 different residue positions, with position 79 unique in having several variants that result in Gq bias (*Figure 3—figure supplement 1D*). Many of the variants with extreme Gq or Gs bias are located within the regions of MC4R that interact with G proteins, with some scattered throughout the transmembrane domains and far fewer in the vicinity of the protein's orthosteric site (*Figure 3B–D*).

## Structural insights into biased signaling

Comparing these results with existing structural information could provide additional detailed insights into MC4R's signaling functions. It is thought that ligand binding in GPCR orthosteric sites is communicated to the intra-cellular G-protein binding domain through a series of conserved residues or 'microswitches' (*Hauser et al., 2021*; *Zhou et al., 2019*). Structural studies comparing the inactive and active state structures have confirmed that MC4R shares a similar signaling cascade (*Yu et al., 2020*; *Zhang et al., 2021*). Upon ligand binding, W258 (W258$^{6x48}$ in https://gpcrdb.org/ nomenclature, where 6 corresponds to the 6th transmembrane helix and 48 denotes 258 is 2 residues before the most conserved residue in that helix, *Isberg et al., 2015*) of the conserved CWxP motif undergoes a conformational rearrangement that is translated to L133$^{3x36}$ and I137$^{3x40}$, of the conserved PIF motif (MIF in melanocortin receptors). This causes F254$^{6x44}$ in the PIF motif to rearrange, which in turn disrupts the packing of three different interactions: (1) L140$^{3x43}$ and I143$^{3x46}$, (2) I251$^{6x41}$ and L247$^{6x37}$, and (3) R147$^{3x50}$ and N240$^{6x30}$. These, amongst other rearrangements, culminate in the receptor being able to bind a G-protein. This interaction with the G-protein is primarily mediated through R147$^{3x50}$ in the conserved DRY motif, T150$^{3x53}$, Y157$^{34x53}$, H158$^{34x54}$, and R305$^{7x56}$ (*Zhang et al., 2021*).

Strikingly, a number of mutations at residues throughout this signaling cascade had extremely positive PC2 values, implicating them as Gq-biasing mutations. Within the core of the cascade, we identified I137$^{3x40}$L, F254$^{6x44}$P, and L140$^{3x43}$I as Gq-biased (*Figure 3A*; *Figure 3—figure supplement 1C*). We also identified a number of Gq-biasing mutations within the G-protein binding pocket, specifically T150$^{3x53}$G and H158$^{34x54}$R (*Figure 3A and C*; *Figure 3—figure supplement 1D*). Interestingly, the H158$^{34x54}$R variant is found in the human population (*Hinney et al., 2006*; *Wade et al., 2021*) and has previously been shown to preferentially signal through the Gq pathway (*Paisdzior et al., 2020*). H158$^{34x54}$ is also co-located near two other mutations (K164$^{3x55}$L, F152$^{3x55}$R), in intracellular loop 2 (ICL2) and near the ends of the third and fourth transmembrane domains (TM3 and TM4), that display bias (*Figure 3C*). Interestingly, K164$^{3x55}$L exhibits a Gs bias in that it drives loss of function through Gq.

Our data also point to a number of potentially novel interactions. For example, M79$^{2x39}$ packs against residues H387$^{G.H5.19}$ and Q390$^{G.H5.22}$ of the Gs alpha subunit (G$\alpha_s$; *Flock et al., 2015*). This position has multiple different variants that result in Gq bias, including M79$^{2x39}$R, M79$^{2x39}$S, and M79$^{2x39}$G (*Figure 3A*, *Figure 3—figure supplement 1D*). The most extreme bias signal in our PCA analysis came from I223$^{5x69}$L (*Figure 3A and D*), which interfaces with the C-terminus of G$\alpha_s$ (*Flock et al., 2015*), near position L394. Further down toward the intracellular side of TM5, we also identified V228R (*Figure 3A and D*), which interfaces proximal to E323$^{G.hgh4.13}$ in G$\alpha_s$ (*Flock et al., 2015*). Collectively, the combination of DMS data and structural information is a fruitful avenue for generating protein structure-function hypotheses. These results highlight the power of DMS to identify regions of MC4R that could be harnessed for designing drugs that precisely modulate specific cellular signaling functions.

## Systematic prediction of treatment response

Many variants of MC4R disrupt signaling by causing protein misfolding, which ultimately inhibits proper localization of MC4R to the cell membrane (*Huang et al., 2017*). Correctors are a class of small molecule drugs that facilitate protein folding. Corrector therapies have been developed for phenotypes such as cystic fibrosis (*Boyle and De Boeck, 2013*) and Fabry disease (*Germain et al., 2016*), and they have been proposed as a strategy for treating *MC4R*-associated obesity (*Huang et al., 2017*; *Huang and Tao, 2014*; *Wang et al., 2014*). One feature of corrector therapies is that they are typically only effective, and therefore FDA-approved, for a subset of patients harboring specific sequence

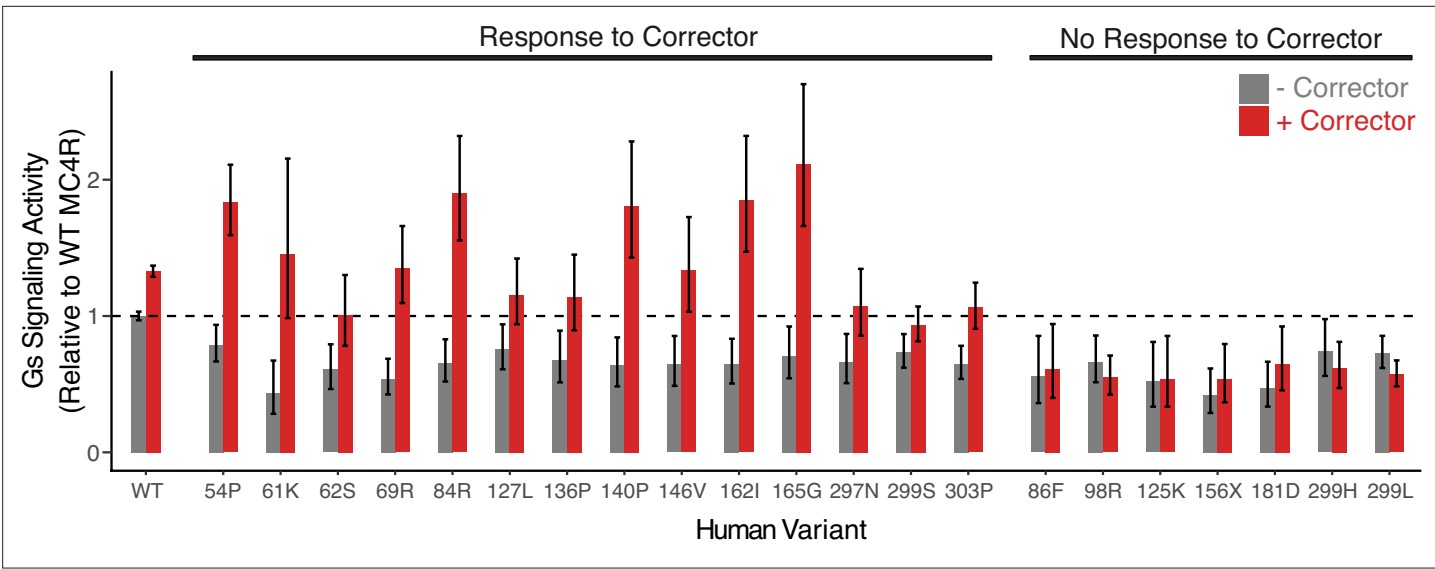

**Figure 4.** Corrector therapy rescues the activity of a subset of human MC4R variants. Gs signaling activity of 21 selected MC4R variant alleles (of 6633 tested) with (red) and without (gray) Ipsen 17, a small molecule corrector that has been shown to restore the activity of misfolded MC4R. Bars represent the activity of the variant allele normalized to that of WT MC4R in the no corrector condition, and error bars are +/-two standard errors.

The online version of this article includes the following figure supplement(s) for figure 4:

**Figure supplement 1.** Identification of variants that respond to corrector treatment.

variants (*Boyle and De Boeck, 2013*; *Weaver et al., 2022*). Identifying variants that respond to corrector therapy is typically done by rigorously testing the effect of a compound on a single variant at a time (*Weaver et al., 2022*), and DMS offers an attractive avenue to systematically test the treatment response of thousands of patient variants in a single assay.

To this end, we tested whether Ipsen 17, a small molecule tool compound that has been shown to correct MC4R misfolding (*Poitout et al., 2007*; *Wang et al., 2014*), is able to restore the Gs signaling function of the MC4R variants in our DMS library. Out of all 6633 tested variants, 290 (4.4%) showed disrupted Gs signaling in the absence of treatment that was partially or fully rescued by the addition of Ipsen 17 (*Figure 4*; *Figure 4—figure supplement 1*, see Materials and methods for details of statistical analysis). This includes a number of variants that have been classified as pathogenic in ClinVar (*Landrum et al., 2014*) or otherwise found in patient sequencing studies (*Brouwers et al., 2021*; *Farooqi et al., 2003*; *Hinney et al., 2013*; *Hinney et al., 2006*; *Huang et al., 2017*; *Stutzmann et al., 2008*; *Wade et al., 2021*; *Yeo et al., 2003*; *Figure 4* shows results for selected variants reported in the human population). Other reported patient variants showed no functional improvement in response to corrector therapy (*Figure 4*). Collectively, these data support that performing DMS in the presence of a small molecule corrector can be used to systematically predict which patients are likely to benefit from such treatment modalities.

## Mapping protein-ligand interactions

DMS experiments can be used to define 'drug-resistant' variants within MC4R that disrupt the activity of different types of ligands, providing functional insight into protein-ligand interactions that are key for understanding the mechanisms underlying agonism. Such functional information would be a valuable addition to structural methods and has the potential to streamline the lengthy and iterative cycle of compound optimization in drug discovery. Substantial work has been done to characterize how peptide agonists interact structurally with MC4R, but similar work on small-molecule agonists with comparable activity and selectivity remains relatively limited (*Gonçalves et al., 2018*; *Heyder et al., 2021*; *Sharma et al., 2019*; *Yu et al., 2020*; *Zhang et al., 2021*). To characterize the functional interaction landscapes of different ligand types and to better understand what distinguishes peptide and small-molecule MC4R pharmacophores, we performed DMS assays of MC4R using both native peptide agonist stimulation (α-MSH) and small molecule agonist stimulation (THIQ). Bayesian

meta-regression analysis of the lowest dose concentrations of α-MSH and THIQ for the Gs signaling assay revealed a set of variants that uniquely disrupt activation by one ligand but not the other (at FDR <5%, *Figure 5A*). These variants cluster exclusively within the orthosteric binding pocket (*Figure 5B–E*; *Figure 5—figure supplement 1*) and at positions of known binding interactions of each molecule (*Zhang et al., 2021*). Notably, there are many more variants that uniquely disrupt activation by α-MSH (*Figure 5A–D*). For example, a majority of amino acid substitutions at position I104$^{2x6}$ disrupt activation of MC4R by α-MSH, but none lead to significant reduction in MC4R activity upon stimulation by THIQ (*Figure 5C*). Multiple positions within the orthosteric binding pocket displayed this pattern (*Figure 5B–E*), which is consistent with how the peptide agonist utilizes a larger network of interactions to increase binding affinity.

Interestingly, this comparison also highlights how the same residue of MC4R can be critical for interfacing with multiple ligands but points to substantive differences in the precise physical interaction between each ligand and that position of the target. For example, positions P48$^{1x36}$ and I129$^{3x32}$ harbor variants that can have differentially deleterious effects under the two ligand conditions (*Figure 5C–E*). Many amino acid substitutions at P48$^{1x36}$ (including to V, I, F, Y, Q, and R) disrupt stimulation of MC4R by α-MSH but not THIQ. However, changing this position to the negatively charged P48$^{1x36}$D variant has the opposite effect, uniquely ablating activation by THIQ. The HFRW motif (His6-Phe7-Arg8-Trp9) of α-MSH represents a conserved pharmacophore critical for activation of MC4R by peptides, and the tri-branched THIQ molecule (R1-R2-R3) mimics the HFRW conformational architecture (*Figure 5E*; *Gonçalves et al., 2018*; *Hruby et al., 1987*; *Zhang et al., 2021*). Our observation that α-MSH stimulation is more sensitive to variants at position P48$^{1x36}$ is consistent with how the His6 of α-MSH forms more interactions with this hydrophobic pocket of MC4R, while the analogous R3 group of THIQ is more flexible and forms non-specific interactions in this region (*Figure 5C–E Gonçalves et al., 2018*; *Hruby et al., 1987*; *Zhang et al., 2021*). At position I129$^{3x32}$, mutation to any polar uncharged variant (S, T, N, or Q) alters THIQ activation, while I129$^{3x23}$V uniquely inhibits α-MSH activation (*Figure 5C–E*). The Phe7 of α-MSH and the analogous R2 group of THIQ form key interactions with Ca$^{2+}$ and the core hydrophobic pocket formed by I129$^{3x32}$ of MC4R (*Figure 5E*; *Zhang et al., 2021*). The enrichment of variants at I129$^{3x32}$ that uniquely disrupt MC4R activation by THIQ points to a stronger dependency of the small-molecule on interactions with this residue. In summary, by assessing the functional consequences of all mutations within the MC4R orthosteric site, we not only confirm known binding interactions but also reveal interactions that distinguish peptide and small-molecule activation. These relationships provide additional functional insight into the structural mechanism of MC4R ligand binding that could be harnessed for drug design.

## Discussion

DMS holds substantial potential for improving many phases of drug discovery and development, but realizing this potential requires the ability to draw highly quantitative and disease-relevant conclusions from DMS data, an outstanding challenge for the field. Here, we demonstrated the value of improved DMS assays and analysis methods in addressing these challenges by applying our approaches to a medically relevant GPCR, MC4R. First, we were able to quantify differences in Gs-mediated cAMP and Gq-mediated calcium signaling for MC4R in response to its native ligand, α-MSH. Understanding these subtle differences in molecular signaling phenotypes is crucial for driving precision treatments. For example, Gq signaling bias is one of the purported mechanisms for the better side-effect profile of the FDA-approved MC4R agonist, Setmelanotide (*Sharma et al., 2019*). Additionally, we systematically assessed the responses of MC4R variants implicated in human obesity to a small-molecule corrector, Ipsen 17. Whereas traditional approaches would have required the separate experimental assessment of every variant, DMS tests every conceivable single amino acid variant in one experiment. This enables researchers to understand how a potential treatment will work for broader swaths of the population before drugs reach the clinic, which could ultimately lead to more effective clinical trial designs.

Our data also contribute to the growing understanding of the complicated mechanism by which MC4R (and other GPCRs) translate ligand binding on the extracellular side of the membrane, through an allosteric network of residues, to G-protein activation on the intracellular side (*Howard et al., 2024*). We demonstrate the utility of DMS for understanding how proteins bind other molecules. In particular, we identified mutations that uniquely disrupt the binding of two agonists, one a peptide

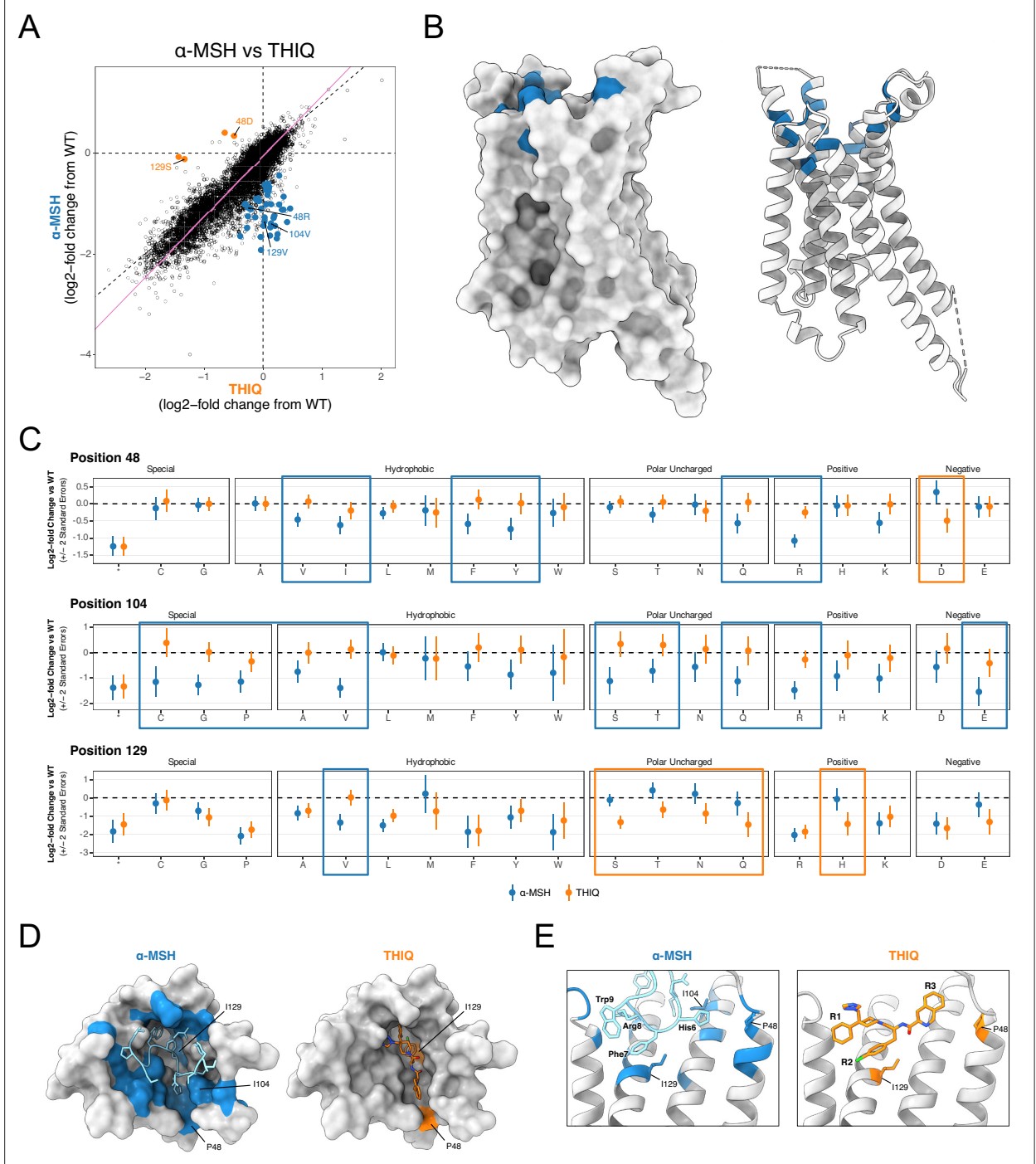

**Figure 5.** Systematic identification of functional protein-ligand interactions. (**A**) Bayesian meta-regression of the α-MSH and THIQ datasets at low agonist concentration for Gs reveals variants that differentially affect MC4R activation by each agonist. Statistically significant effects are colored by which agonist condition they impaired (α-MSH, blue; THIQ, orange, FDR <0.05). Variants at positions 48, 104, and 129 are labeled. (**B**) Side-views of MC4R structure in surface (left) and ribbon (right) view show clustering of variants from A that specifically abrogate α-MSH but not THIQ activation (blue) in the extracellular orthosteric binding site. (**C**) Effect of all possible variants at residues 48, 104, and 129 at low concentration of α-MSH (blue) and THIQ (orange). Variants that disproportionately affect activation by only one agonist are boxed by respective color. Error bars are +/-2 standard errors. (**D**) Top-down surface views of MC4R (PDB: 7f58) with bound α-MSH (left, blue; PDB: 7f53) or THIQ (right, orange; PDB: 7f58). Positions are colored by whether variants at that position uniquely perturb activation by α-MSH (blue) or THIQ (orange). (**E**) Zoomed view of the MC4R binding pocket with α-MSH bound (left) or THIQ bound (right). Structures are colored as in (**D**). Residues that form the HFRW motif of α-MSH and functional groups R1, R2, and R3 of THIQ are labeled in bold. Residues 48, 104, and 129 are shown in stick form.

*Figure 5 continued on next page*

*Figure 5 continued*

The online version of this article includes the following figure supplement(s) for figure 5:

**Figure supplement 1.** Variants that differentially impact activation by peptide versus small molecule agonists.

(α-MSH) and one a small molecule (THIQ). This is crucial information for understanding the mechanism by which these molecules bind MC4R, which could lead to hypotheses for further optimization of more potent compounds. DMS data contain a substantial amount of latent information about protein structure, and can help identify novel pockets to target with new compounds (*Weng et al., 2024*). As suggested above, we also envision using DMS data to help optimize molecules by identifying mutations that uniquely disrupt or potentiate sets of compounds in a chemical series.

Importantly, the experimental and analysis frameworks we describe are widely applicable to GPCRs and more broadly to other target classes. GPCRs are known to signal through a variety of pathways, and there are existing transcriptional reporters for most G-protein-mediated signaling pathways that could be harnessed to gain a holistic understanding of the downstream consequences of GPCR activity (*Hauser et al., 2022*). Furthermore, there are a substantial number of existing transcriptional reporters for a wide variety of cellular signaling processes mediated by transcription factors, kinases, and other major targets of drug discovery efforts, which could extend these methods to other target classes.

Finally, there has been incredible progress over the last few years in the development of machine learning-based models to predict the structural and functional consequences of mutations on proteins. That said, these models have struggled to address many real-world applications in the treatment of human disease (*McDonald et al., 2024*). One potential explanation to this gap in performance could be due to the fact that many DMS datasets (our previous work included) used to train these models typically assay a limited set of experimental conditions, often assay cellular effects that are only indirectly related to disease etiology, suffer from low signal-to-noise ratios, have no quantification of uncertainty in their measurements, and are subject to a heterogeneous set of subtle experimental caveats. Detailed experimental characterization of protein function (such as the >21,500,000 different measurements in this work alone) and efforts such as ProteinGym (*Notin et al., 2023*) to benchmark large-scale functional data will continue to be critical for developing and evaluating large scale models that can be applied to a wider variety of drug discovery and development applications. As both the experimental and computational methods continue to improve, we foresee DMS having a profound impact on the drug discovery process.

## Materials and methods
### Reporter assays and cell line development

The Gs (cAMP-luciferase) reporter assay, diagramed in *Figure 1—figure supplement 1A*, was adapted from an assay previously used to assess the function of other GPCRs (*Jones et al., 2020*; *Jones et al., 2019*).

The Gq assay relay reporter system (diagrammed in *Figure 1—figure supplement 1B*) is described in detail in a patent stemming from this work (*Chan et al., 2024*). Briefly, it was constructed as follows: A piggybac transposon plasmid was constructed using Gibson assembly harboring a genetic cassette that expresses a synthetic transcription factor, Gal4_DBD-VPR under the control of an NFAT response element. Human embryonic kidney cells (HEK293T) were then cotransfected with this plasmid along with a piggybac transposase expression vector, and cells were selected for puromycin resistance. These cells were then isocloned by dilution plating, and colonies were selected based on single cell clones. Five clones were then carried forward based on cell morphology and growth rates. A second series of Bxb1-based landing pad vectors were constructed containing a library of 20 Gq-coupled GPCRs under the control of the dox-inducible promoter, along with a second genetic cassette containing a DNA-barcoded luciferase gene under the control of the Gal4_UAS enhancer. This plasmid library was integrated into each of the five isoclonal Gq-relay cell lines. The best performing Gq-relay cell line was selected on signal-to-noise criteria across a panel of agonists for the 20 Gq-coupled GPCRs. Use of this relay system in combination specifically with an *MC4R* transgene resulted in α-MSH-dose dependent expression of the reporter gene (*Figure 1—figure supplement 1C*).

## Building the DMS libraries

Generation and genomic integration of the plasmid libraries containing all possible single amino acid substitutions of MC4R used a further optimized version of an earlier method (*Jones et al., 2020*). In brief, variant segments of *MC4R* cDNA were amplified from DNA microarrays and cloned into base vectors through a multi-step process to yield pooled libraries of *MC4R* variants with fully intact and barcoded reporter gene cassettes. Fully assembled plasmid libraries were then co-transfected with a plasmid encoding Bxb1 recombinase into HEK293T cells containing a landing pad at the H11 safe harbor locus to achieve single copy integration per cell. In a deviation from the previously published method, two independent replicates of each sub-library were cloned and pooled together post-cellular integration in order to maximize library coverage and the number of barcodes per variant.

## Variant-barcode mapping

As described above, barcodes were randomly appended to variants during amplification of MC4R segments from variant oligo pools and then ligated into library base vectors. After this first step of plasmid library cloning, variant segments were amplified along with the neighboring barcode and sequenced with 2x150 paired-end reads (see *Supplementary file 1c* for amplification and sequencing primers) on an Illumina NextSeq 550 instrument using a 300-cycle Mid Output kit. Illumina 2x150 BCL files were demultiplexed with *bcl2fastq2* into R1 and R2 FASTQ files, which were merged into single fragments using *Flash2* requiring a minimum 5 bp of overlap (*Magoč and Salzberg, 2011*). The first 21 bp of each fragment corresponding to the barcode sequence were extracted into the read name, and the remaining fragment was adapter-trimmed using *umi_tools* and *cutadapt*, respectively (*Martin, 2011*; *Smith et al., 2017*). The remaining fragments were mapped against a custom reference composed of the designed oligonucleotide library using *STAR* with default parameters except requiring that alignments be strictly unique to be reported (*Dobin et al., 2013*). Taking each alignment as an oligo-barcode pair, the read counts per unique oligo-barcode pair were computed for each replicate and joined by barcode. Finally, the resulting maps were filtered to require each oligo-barcode pair to pass three requirements in both replicates: correct barcode length, total read depth >10, and purity >0.75. The purity of an oligo-barcode pair was defined as the read count of that pair divided by the total number of reads containing that barcode. Post-processing after *STAR* was performed using *samtools* for BAM manipulation and custom R code (*Li et al., 2009*).

## Running DMS assays

The DMS assay protocol was optimized from a previous method (*Jones et al., 2020*). HEK293T single-copy variant cell libraries were seeded at a density of ~17 × $10^6$ cells per 150 mm tissue-culture treated dish in DMEM +10% fetal bovine serum (FBS). Four dishes were seeded for each experimental condition, with each dish being treated as an independent biological replicate (four replicates per condition). Twenty-four hours after seeding, media was exchanged with DMEM +0.5% FBS+/-10 ng/mL Doxycycline. For chaperone experiments, all conditions were additionally replicated +/-1 µM Ipsen 17. 24 hrs after Doxycycline induction, media was exchanged with Opti-MEM +DMSO, Forskolin, or MC4R agonist (α-MSH or THIQ). Forskolin bypasses MC4R to constitutively activate cAMP signaling, so this condition was used as a variant-independent measurement of library composition. For chaperone experiments, cells were washed 3 x with 10 mL DMEM to remove Ipsen 17 prior to agonist stimulation as it has been shown to be an antagonist of α-MSH activity and is thought to bind directly to the same site on MC4R (*Wang et al., 2014*). Six hours after agonist stimulation, cells were harvested by scraping in 4 mL lysis buffer (RLT buffer [QIAGEN] +143 mM β-ME). Given the seeding density (~17 × $10^6$ cells per 150 mm replicate dish), time from seeding to collection, and doubling time of HEK293T cells, approximately 25.5x$10^6$ cells were collected per replicate. This translates to approximately 30–60 x cellular coverage per amino acid variant in each replicate. Lysis was performed by passing the cell slurry 6 x through a sterile 18 G needle and then spinning through QIAshredder (QIAGEN) columns. RNA was extracted from 1 ml of the homogenized lysate with the RNeasy Plus Mini kit (QIAGEN), including optional on-column DNAse digestion, and eluted into 100 µL water. Eight reverse transcriptase reactions per sample were performed with the SuperScript IV kit (Thermo Fisher), as described previously (*Jones et al., 2020*) (primers listed in *Supplementary file 1c*). cDNA from each sample was treated with 1 µL RNase A (100 µg/ml, Thermo Fisher) and 3.2 µL RNase H (5000 U/mL, NEB) at 37 °C for 30 min. RNase-treated samples were concentrated to ~55 µL by spinning

through Amicon Ultra 10 kDa concentrators (EMD Millipore) for ~8 min. To determine the necessary cycle numbers for equivalent amplification of each sample library, qPCR reactions were performed on 1 µL cDNA (diluted 1:8 in water) with Q5 polymerase (NEB), SYBR Green (Thermo Fisher), and library amplification primers (**Supplementary file 1c**). Final amplification cycles for each sample were chosen by adding three cycles to the Cq values generated from each respective qPCR reaction. Illumina sequencing libraries were prepared by amplifying 50 µL of each cDNA sample with sequencing adapters (500 nM each library amplification primer, **Supplementary file 1c**) using the NEBNext Q5 High Fidelity 2 x PCR Kit (NEB) under the following cycling conditions: 98 °C for 30 s, X cycles of 98 °C for 8 s, 65 °C for 20 s, and 72 °C for 10 s, followed by an extension of 72 °C for 2 min. 3 µL of each DNA library sample was run on a 4% E-Gel (Thermo Fisher) and densitometry was performed with Fiji to account for differences in library yields. Samples were mixed at equal amounts into a single pool and then purified into 200 µL IDTE (QIAGEN) with AxyPrep Magnetic beads (Thermo Fisher Scientific). The purified library was quantified with the DeNovix High-Sensitivity Fluorescence kit and prepared for sequencing with a 10% PhiX spike-in. Final library mixture was sequenced using custom read and index primers **Supplementary file 1c** on an Illumina NextSeq 550 with the High Output 75 cycle kit.

## Sequence processing for barcode expression

Illumina 1x26 BCL files were demultiplexed with *bcl2fastq2* and processed to remove the last 5 bp using basic bash commands. The resulting sequences were counted for each sample, and the resulting barcodes were joined with the appropriate oligo-barcode map. The resulting barcodes were joined with sample and MC4R variant metadata and returned for regression analysis. All processing after demultiplexing was performed with custom R code. Total mapped reads per replicate at the RNA-seq stage were as follows:

- Gs/CRE: 9.1–18.2 million mapped reads, median = 12.3
- Gq/UAS: 8.6–24.1 million mapped reads, median = 14.5
- Gs/CRE +Chaperone: 6.4–9.5 million mapped reads, median = 7.5

The median read counts per sample per barcode were 8, 10, and 6 reads for Gs/CRE, Gq/UAS, and Gs/CRE + Chaperone assays, respectively. The median number of barcodes per variant across all samples (the 'median of medians') were 56 for Gs/CRE, 28 for Gq/UAS, and 44 for Gs/CRE + Chaperone. The correlation (r) of barcode readcounts between replicates was ~0.5 and~0.4 for the Gs and Gq assays, respectively (**Figure 1—figure supplement 1E**).

## Negative binomial regression analysis pipeline

In many DMS methods, barcodes are summed within protein-coding variants to generate a single variant-level count per sample. This unnecessarily sacrifices available power obtained by repeatedly measuring the same molecular process in distinct cells. An alternative is to model barcode counts directly. However, to apply standard models, missing data must either be removed, which can remove a majority of detected barcodes, or somehow imputed. Additionally, many existing methods either use a log transformation of read counts to obtain approximate normality, or apply Poisson regression and related assumptions for inference (**Faure et al., 2020**; **Rubin et al., 2017**). However, there is strong prior reason to believe that DMS counts are overdispersed in our data, since our synthetic reporter system is read out via RNA-seq (**Robinson and Smyth, 2007**).

Instead, we developed and applied a mixed effects negative binomial general linear model (GLM) to resolve these challenges. These models have been widely deployed to model count data, and in particular identify differential expression, in bulk and single-cell RNA-seq analysis. We implement maximum likelihood estimation for this model using *glmmTMB*, which can accommodate the potentially large scale of multiplex count data (**Brooks et al., 2017**). For each position, we consider all variants located at that position along with all wild-type variants in the same protein subregion and apply the following model:

$$
\begin{aligned}
\text{count}_{ijkm} &\sim \text{NB}(\mu_{ij}, \theta) \\
\alpha_k &\sim \mathcal{N}(0, \sigma^2) \\
\log(\mu_{ij}) &= \beta_i^{\text{condition}} + \beta_{ij}^{\text{variant}} + \alpha_k + \text{offset}(\gamma_m)
\end{aligned}
$$

For the $i$th condition, the $j$th variant, the $k$th barcode, and the $m$th sample. Consequently, the first two terms in the last equation above correspond to a global mean term for each condition and a term for the variant-specific deviation from wild-type in each condition. The last two terms are the random effect for barcode $k$, and the sample-specific technical offset for sample $m$. The definition of the offset is often context specific, and here we use the log of the sum of barcode counts derived from stops, reasoning they should be constant across conditions and replicates.

We fit this model for each MC4R position independently and extract coefficients for the additive shift in the mean of each variant relative to wild-type. Using the per-condition summary statistics, we obtained Wald test statistics by dividing the effect size by the standard error and computed p-values against the normal distribution. p-values were adjusted for multiple testing using the Benjamini-Hochberg method and thresholded to 1% or 5% FDR where indicated.

To define more complex null hypotheses like chaperone rescue, we extracted marginal means for each variant under each treatment using the *emmeans* package (*Lenth, 2024*). We define the chaperone rescue contrast as the additive shift of each variant in each treatment condition to the wild-type mean specifically in the untreated condition. Since this quantity is a linear contrast across marginal means, we computed the associated contrast estimates and standard errors, and tested them for significance using the same approach as the per-condition summary statistics.

## Comparison to human genetics data and variant effect predictors

Pathogenicity classifications of *MC4R* missense and nonsense variants were obtained from ClinVar (*Landrum et al., 2014*) on January 5, 2024, and all available annotations were included in the analysis regardless of ClinVar review status metric. Human population frequency data for *MC4R* missense variants were obtained from gnomAD (*Metzger et al., 2024*) on January 8, 2024. A comprehensive list of 220 *MC4R* variants that are of potential clinical relevance (*Supplementary file 1b*) was mined from ClinVar (*Landrum et al., 2014*), along with papers or review articles describing variants identified in human sequencing studies (*Brouwers et al., 2021*; *Farooqi et al., 2003*; *Hinney et al., 2013*; *Hinney et al., 2006*; *Huang et al., 2017*; *Rodríguez Rondón et al., 2024*; *Stutzmann et al., 2008*; *Wade et al., 2021*; *Yeo et al., 2003*). Effect predictions for *MC4R* missense variants were obtained from the public releases of AlphaMissense (*Cheng et al., 2023*) and popEVE (*Orenbuch et al., 2023*).

## Identification of functionally biased variants

We used Principal Component Analysis to identify biased variants. Specifically, we used the test statistic ($\log_2$ fold change divided by the standard error) for the DMSO and all α-MSH conditions in both the Gq and Gs pathways to create a matrix with conditions (defined as Drug_Concentration) as columns, and each individual mutant (defined as Position_Substitution) as rows. We then passed this matrix to R's *prcomp* function with default parameters, and used the first two principal components to visualize the results. Visual inspection of the loadings via a biplot, along with plotting of various variant types (*Figure 3—figure supplement 1A and B*) revealed that PC1 separates variants based on their overall effect on MC4R function, and PC2 separates variants based on differential response through the Gq and Gs pathways. We set a simple cutoff of +/- 7.5 on PC2 to highlight particularly interesting variants.

## Identifying variants that respond to corrector treatment

We identified variants whose activity increased upon treatment with Ipsen 17 using a slight modification of the summary statistics from the general-purpose model described above. For each variant and condition, we compute the $\log_2$-scaled marginal mean (averaging across barcodes and replicates) and its associated standard error. Then, for each variant, we compute the following two summary statistics, and test whether their estimates are significantly different from zero. First, we define a variants' *defect* as the marginal mean of that variant under DMSO treatment minus the marginal mean of WT under DMSO treatment. This quantifies the existence and severity of the variant's intrinsic effect on MC4R activity. Second, we define that variant's *rescue* as the marginal mean of that variant under Ipsen 17 treatment minus the marginal mean of that variant under DMSO activity. This quantifies the magnitude of the (typically) increase in MC4R activity upon Ipsen 17 treatment for each variant, relative to its DMSO-only baseline. After computing the indicated estimate and errors, we perform significance testing as in the general model, where we define Wald statistics as the estimate divided by

the propagated error, compute p-values from the normal distribution, and adjust for multiple testing using the Benjamini-Hochberg procedure.

## Identifying critical variants for protein-ligand interactions

We identified sets of mutations that specifically inhibited or potentiated MC4R activation in the presence of ligands. We applied the general DMS model (see *Negative binomial regression analysis pipeline*) and extracted $log_2$-scaled fold changes and standard errors for each variant relative to WT, within each ligand-treated condition. To account for systematic differences between ligands across all variants, we applied Bayesian meta-regression via the *brms* R package (*Bürkner, 2017*) and regressed the α-MSH summary statistics against THIQ, while including the errors in both quantities via the se() and me() brms functions. Finally, to infer significant α-MSH- or THIQ-specific effects, we extracted the residual of each variant relative to the meta-regression best-fit line and tested whether this residual was significantly non-zero based on the posterior sampling performed with *brms*.

## Structural modeling

Molecular visualization of variant effects on MC4R was performed with UCSF ChimeraX (*Pettersen et al., 2021*). For visualization of functionally biased extremes in MC4R (*Figure 3B*), the maximum absolute PC2 value for each position was calculated and the define attribute function of ChimeraX was used to color the structure of α-MSH-bound MC4R (PDB: 7F53) by these relative values, ranging from white (no bias) to pink (extreme bias). For all other structural panels related to variant bias, positions of interest are colored binarily by green or purple to indicate Gs-bias or Gq-bias, respectively. For visualizing variant effects on protein-ligand interactions (*Figure 5*), positions with significant variants identified by meta-regression were colored by whether variants at a given position perturb activation uniquely by α-MSH (blue) or THIQ (orange). Where α-MSH and THIQ structures are shown, the respective α-MSH-bound (PDB: 7F53) and THIQ-bound (PDB: 7F58) cryo-EM models were used for visualization. For the depictions in *Figure 5B*, α-MSH-bound MC4R was used.

## Cell line validation

STR profiling performed by the University of California Berkeley DNA Sequencing Facility confirmed that all cell lines used were HEK293-derived. Cell lines were verified as negative for mycoplasma contamination using the EZ-PCR Mycoplasma Detection Kit (Sartorius).

# Acknowledgements

We thank Jakob Sture Madsen for his intellectual contributions to the project and his helpful comments on the paper.

# Additional information

### Competing interests

Conor J Howard, Nathan S Abell, Beatriz A Osuna, Dean R Artis, Jonathan B Asfaha, Joshua S Bloom, Andrew Liao, Eden Mahdavi, Nabil Mohammed, Alan L Su, Giselle A Uribe, Diane E Dickel, Nathan B Lubock: Current or past employee of Octant, Inc and/or hold shares or options in the company. Eric M Jones, Leon Y Chan, Henry Chan, Aaron R Cooper, Sriram Kosuri: Current or past employee of Octant, Inc and/or hold shares or options in the company. Inventor of patent related to this work.

### Funding

| Funder | Grant reference number | Author |
| --- | --- | --- |
| National Institute of General Medical Sciences | R43GM137745 | Sriram Kosuri |

The funders had no role in study design, data collection and interpretation, or the decision to submit the work for publication.

## Author contributions

Conor J Howard, Conceptualization, Data curation, Formal analysis, Validation, Investigation, Visualization, Methodology, Writing – original draft, Writing – review and editing; Nathan S Abell, Conceptualization, Data curation, Software, Formal analysis, Validation, Investigation, Visualization, Methodology, Writing – original draft, Writing – review and editing; Beatriz A Osuna, Eric M Jones, Leon Y Chan, Henry Chan, Dean R Artis, Joshua S Bloom, Aaron R Cooper, Andrew Liao, Eden Mahdavi, Alan L Su, Giselle A Uribe, Conceptualization, Data curation, Formal analysis, Validation, Investigation, Visualization, Methodology, Writing – review and editing; Jonathan B Asfaha, Conceptualization, Resources, Data curation, Formal analysis, Validation, Investigation, Visualization, Methodology, Writing – review and editing; Nabil Mohammed, Conceptualization, Data curation, Software, Formal analysis, Validation, Investigation, Visualization, Methodology, Writing – review and editing; Sriram Kosuri, Conceptualization, Supervision, Funding acquisition, Methodology, Project administration, Writing – review and editing; Diane E Dickel, Conceptualization, Data curation, Formal analysis, Supervision, Validation, Investigation, Visualization, Methodology, Writing – original draft, Project administration, Writing – review and editing; Nathan B Lubock, Conceptualization, Data curation, Software, Formal analysis, Supervision, Validation, Investigation, Visualization, Methodology, Writing – original draft, Project administration, Writing – review and editing

## Author ORCIDs

Conor J Howard ⓘ https://orcid.org/0000-0001-5375-6248
Beatriz A Osuna ⓘ https://orcid.org/0000-0003-2604-6173
Eric M Jones ⓘ https://orcid.org/0000-0002-6648-1965
Leon Y Chan ⓘ https://orcid.org/0000-0002-0189-4689
Joshua S Bloom ⓘ https://orcid.org/0000-0002-7241-1648
Sriram Kosuri ⓘ https://orcid.org/0000-0002-4661-0600
Diane E Dickel ⓘ https://orcid.org/0000-0001-5497-6824
Nathan B Lubock ⓘ https://orcid.org/0000-0001-8064-2465

Reviewer #1 (Public review): https://doi.org/10.7554/eLife.104725.3.sa1
Reviewer #2 (Public review): https://doi.org/10.7554/eLife.104725.3.sa2
Author response https://doi.org/10.7554/eLife.104725.3.sa3

---

# Additional files

## Supplementary files

Supplementary file 1. Supplementary tables. (a) Experimental conditions across all DMS assays. (b) Functional effects of >200 human MC4R variants. (c) Oligo sequences used in this study.

MDAR checklist

## Data availability

Raw sequencing data are available from SRA under project accession number PRJNA1161152. All code used for analysis and figure generation are available at GitHub (copy archived at *Abell et al., 2024*).

The following dataset was generated:

| Author(s) | Year | Dataset title | Dataset URL | Database and Identifier |
|---|---|---|---|---|
| Howard CJ, Abell NS, Osuna BA, Jones EM, Chan LY, Chan H, Artis DR, Asfaha JB, Bloom JS, Cooper AR, Liao A, Mahdavi E, Mohammed N, Uribe GA, Kosuri S, Dickel DE, Lubock NB | 2024 | High resolution deep mutational scanning enables target characterization for drug discovery | https://www.ncbi.nlm.nih.gov/bioproject/?term=PRJNA1161152 | NCBI BioProject, PRJNA1161152 |

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

## Appendix 1

**Appendix 1—key resources table**

| Reagent type (species) or resource | Designation | Source or reference | Identifiers | Additional information |
|---|---|---|---|---|
| gene (*H. sapien*) | MC4R | | Uniprot:P32245 | |
| genetic reagent | Gs luciferase | this study | | |
| genetic reagent | Gq relay | this study | | |
| cell line (*H. sapien*) | HEK293T | ATCC | | |
| chemical compound, drug | Ipsen 17 | WuXi AppTec | CAS:852803-39-5 | Custom synthesis |
| chemical compound, drug | Forskolin | Sigma Aldrich | CAS:66575-29-9 | |
| chemical compound, drug | a-MSH | ApexBio | CAS:581-05-5 | |
| chemical compound, drug | THIQ | MedChemExpress | CAS:312637-48-2 | |
| software, algorithm | umi_tools | *Smith et al., 2017* | | |
| software, algorithm | cutadapt | *Martin, 2011* | | |
| software, algorithm | STAR | *Dobin et al., 2013* | | |
| software, algorithm | samtools | *Li et al., 2009* | | |
| software, algorithm | FLASH2 | *Magoč and Salzberg, 2011* | | |

