## [Editor Report · eLife Assessment]

The authors use deep mutational scanning to assess the effect of ~6,600 protein-coding variants in MC4R, a G-protein-coupled receptor associated with obesity. They develop new, more precise approaches to deep mutational scanning, enabling them to probe molecular phenotypes directly relevant to the development of drugs that target this receptor. In this **important** work, the authors provide **compelling** evidence that variants impact signaling through MC4R in different ways, that some defective variants are amenable to a corrector drug and that deep mutational scanning data could guide compound optimization.

---

## [Referee Report · Reviewer #1 (Public review)]

Summary:

Howard et al. performed deep mutational scanning on the MC4R gene, using a reporter assay to investigate two distinct downstream pathways across multiple experimental conditions. They validated their findings with ClinVar data and previous studies. Additionally, they provided insights into the application of DMS results for personalized drug therapy and differential ligand responses across variant types.

Strengths:

They captured over 99% of variants with robust signals and investigated subtle functionalities, such as pathway-specific activities and interactions with different ligands, by refining both the experimental design and analytical methods.

They provided additional details regarding the quality of the library, including the even composition of variants, sufficient readout from tested cells, and adequate sequencing depth. Additionally, they clarified the underlying assay mechanisms, effectively demonstrating the robustness of their results.

---

## [Referee Report · Reviewer #2 (Public review)]

Overview

In this manuscript the authors use deep mutational scanning to assess the effect of ~6,600 protein-coding variants in MC4R, a G protein-coupled receptor associated with obesity. Reasoning that current deep mutational scanning approaches are insufficiently precise for some drug development applications, they focus on articulating new, more precise approaches. These approaches, which include a new statistical model and innovative reporter assay, enable them to probe molecular phenotypes directly relevant to the development of drugs that target this receptor with high precision and statistical rigor.

They use the resulting data for a variety of purposes, including probing the relationship between MC4R's sequence and structure, analyzing the effect of clinically important variants, identifying variants that disrupt downstream MC4R signaling via one but not both pathways, identifying loss of function variants are amenable to a corrector drug and exploring how deep mutational scanning data could guide small molecule drug optimization.

Strengths

The analysis and statistical framework developed by the authors represent a significant advance. In particular, it makes use of barcode-level internally replicated measurements to more accurately estimate measurement noise.

The framework allows variant effects to be compared across experimental conditions, a task which is currently hard to do with rigor. Thus, this framework will be applicable to a large number of existing and future deep mutational scanning experiments.

The authors refine their existing barcode transcription-based assay for GPCR signaling, and develop a clever "relay" new reporter system to boost signaling in a particular pathway. They show that these reporters can be used to measure both gain of function and loss of function effects, which many deep mutational scanning approaches cannot do.

The use of systematic approaches to integrate and then interrogate high-dimensional deep mutational scanning data is a big strength. For example, the authors applied PCA to the variant effect results from reporters for two different MC4R signaling pathways and were able to discover variants that biased signaling through one or the other pathway. This approach paves the way for analyses of higher dimensional deep mutational scans.

The authors use the deep mutational scanning data they collect to map how different variants impact small molecule agonists activate MC4R signaling. This is an exciting idea because developing small-molecule protein-targeting therapeutics is difficult, and this manuscript suggests a new way to map small molecule-protein interactions.

Weaknesses

The authors derive insights into the relationship between MC4R signaling through different pathways and its structure. While these make sense based on what is already known, the manuscript would be stronger if some of these insights were validated using methods other than deep mutational scanning.

Likewise, the authors use their data to identify positions where variants disrupt MC4R activation by one small molecule agonist but not another. They hypothesize these effects point to positions that are more or less important for the binding of different small molecule agonists. The manuscript would be stronger if some of these insights were explored further.

Impact

In this manuscript the authors present new methods, including a statistical framework for analyzing deep mutational scanning data that will have a broad impact. They also generate MC4R variant effect data that is of interest to the GPCR community.

Comments on revisions:

I do not have additional comments, and feel that the authors addressed most of my concerns!

---

## [Author Response]

The following is the authors’ response to the original reviews.

**Reviewer 1 (Public reviews):**
SummaryHoward et al. performed deep mutational scanning on the MC4R gene, using a reporter assay to investigate two distinct downstream pathways across multiple experimental conditions. They validated their findings with ClinVar data and previous studies. Additionally, they provided insights into the application of DMS results for personalized drug therapy and differential ligand responses across variant types.StrengthsThey captured over 99% of variants with robust signals and investigated subtle functionalities, such as pathway-specific activities and interactions with different ligands, by refining both the experimental design and analytical methods.WeaknessesWhile the study generated informative results, it lacks a detailed explanation regarding the input library, replicate correlation, and sequencing depth for a given number of cells. Additionally, there are several questions that it would be helpful for authors to clarify.(1) It would be helpful to clarify the information regarding the quality of the input library and experimental replicates. Are variants evenly represented in the library? Additionally, have the authors considered using long-read sequencing to confirm the presence of a single intended variant per construct? Finally, could the authors provide details on the correlation between experimental replicates under each condition?Are variants evenly represented in the library?

We strive to achieve as evenly balanced library as possible at every stage of the DMS process (e.g., initial cloning in *E. coli* through integration into human cells). Below is a representative plot showing the number of barcodes per amino acid variant at each position in a given ~60 amino acid subregion of MC4R, which highlights how evenly variants are represented at the *E. coli* cloning stage.

We also make similar measurements after the library is integrated into HEK293T cell lines, and see similarly even coverage across all variants, as shown in the plot below:

**Author response image 2. sa3fig2:** 

Additionally, have the authors considered using long-read sequencing to confirm the presence of a single intended variant per construct?

We agree long-read sequencing would be an excellent way to confirm that our constructs contain a single intended variant. However, we elected for an alternate method (outlined in more detail in Jones *et al.* 2020) that leverages multiple layers of validation. First, the oligo chip-synthesized portions of the protein containing the variants are cloned into a sequence-verified plasmid backbone, which greatly decreases the chances of spuriously generating a mutation in a different portion of the protein. We then sequence both the oligo portion and random barcode using overlapping paired end reads during barcode mapping to avoid sequencing errors and to help detect DNA synthesis errors. At this stage, we computationally reject any constructs that have more than one variant. Given this, the vast majority of remaining unintended variants would come from somatic mutations introduced by the *E. coli* cloning or replication process, which should be low frequency. We have used our in-house full plasmid sequencing method, OCTOPUS, to sample and spot check this for several other DMS libraries we have generated using the same cloning methods. We have found variants in the plasmid backbone in only ~1% of plasmids in these libraries. Our statistical model also helps correct for this by accounting for barcode-specific variation. Finally we believe this provides further motivation for having multiple barcodes per variant, which dilutes the effect of any unintended additional variants.

**Finally, could the authors provide details on the correlation between experimental replicates under each condition?**

Certainly! In general, the Gs reporter had higher correlation between replicates than the Gq system (r ~ 0.5 vs r ~ 0.4). The plots below, which have been added as a panel to Supplementary Figure 1, show two representative correlations at the RNA-seq stage of read counts for barcodes between the low a-MSH conditions.

We added the following text to reference this panel:

(see Methods > Sequence processing for barcode expression): “The correlation (r) of barcode readcounts between replicates was ~0.5 and ~0.4 for the Gs and Gq assays, respectively (Supplementary Fig. 1E).”

One important advantage of our statistical model is that it’s able to leverage information from barcodes regardless of the number of replicates they appear in.

(2) Since the functional readout of variants is conducted through RNA sequencing, it seems crucial to sequence a sufficient number of cells with adequate sequencing saturation. Could the authors clarify the coverage depth used for each RNA-seq experiment and how this depth was determined? Additionally, how many cells were sequenced in each experiment?

The text has been added in the manuscript as follows:

(in Methods > Running DMS Assays): “Given the seeding density (~17x10^6^ cells per 150 mm replicate dish), time from seeding to collection, and doubling time of HEK293T cells, approximately 25.5x10^6^ cells were collected per replicate. This translates to approximately 30-60x cellular coverage per amino acid variant in each replicate.”

(in Methods > Sequence processing for barcode expression): “Total mapped reads per replicate at the RNA-seq stage were as follows:

- Gs/CRE: 9.1-18.2 million mapped reads, median=12.3

- Gq/UAS: 8.6-24.1 million mapped reads, median=14.5

- Gs/CRE+Chaperone: 6.4-9.5 million mapped reads, median=7.5”

The median read counts per sample per barcode were 8, 10, and 6 reads for Gs/CRE, Gq/UAS, and Gs/CRE+Chaperone assays, respectively. The median number of barcodes per variant across all samples (the “median of medians”) were 56 for Gs/CRE, 28 for Gq/UAS, and 44 for Gs/CRE+Chaperone.”

(3) It appears that the frequencies of individual RNA-seq barcode variants were used as a proxy for MC4R activity. Would it be important to also normalize for heterogeneity in RNA-seq coverage across different cells in the experiment? Variability in cell representation (i.e., the distribution of variants across cells) could lead to misinterpretation of variant effects. For example, suppose barcode_a1 represents variant A and barcode_b1 represents variant B. If the RNA-seq results show 6 reads for barcode_a1 and 7 reads for barcode_b1, it might initially appear that both variants have similar effect sizes. However, if these reads correspond to 6 separate cells each containing 1 copy of barcode_a1, and only 1 cell containing 7 copies of barcode_b1, the interpretation changes significantly. Additionally, if certain variants occupy a larger proportion of the cell population, they are more likely to be overrepresented in RNA sequencing.

We account for this heterogeneity in several ways. First, as shown above (see Response to Reviewer 1, Question 1), we aim to have even representation of variants within our libraries. Second, we utilize compositional control conditions like forskolin or unstimulated conditions to obtain treatment-independent measurements of barcode abundance and, consequently, of mutant-vs-WT effects that are due to compositional rather than biological variability. We expect that variability observed under these controls is due to subtle effects of molecular cloning, gene expression, and stochasticity. Using these controls, we observe that mutant-vs-WT effects are generally close to zero in these normalization conditions (e.g., in untreated Gq, see Supplementary Figure 3) as compared to treated conditions. For example, pre-mature stops behave similar to WT in normalization conditions. This indicates that mutant abundance is relatively homogenous. Where there are barcode-dependent effects on abundance, we can use information from these conditions to normalize that effect. Finally, our mixed-effect model accounts for barcode-specific deviations from the expected mutant effect (e.g., a “high count” barcode consistently being high relative to the mean).

(4) Although the assay system appears to effectively represent MC4R functionality at the molecular level, we are curious about the potential disparity between the DMS score system and physiological relevance. How do variants reported in gnomAD distribute within the DMS scoring system?

Figure 2D shows DMS scores (variant effect on Gs signaling) relative to human population frequency for all MC4R variants reported in gnomAD as of January 8, 2024.

(5) To measure Gq signaling, the authors used the GAL4-VPR relay system. Is there additional experimental data to support that this relay system accurately represents Gq signaling?

The full Gq reporter uses an NFAT response element from the IL-2 promoter to regulate the expression of the GAL4-VPR relay. In this system, the activation of Gq signaling results in the activation of the NFAT response element, and this signal is then amplified by the GAL4-VPR relay. The NFAT response element has been previously well-validated to respond to the activation of Gq signaling (e.g., Boss, Talpade, and Murphy 1996). We will have added this reference to the text (see Results> Assays for disease-relevant mechanisms) to further support the use of the Gq assay.

(6) Identifying the variants responsive to the corrector was impressive. However, we are curious about how the authors confirmed that the restoration of MC4R activity was due to the correction of the MC4R protein itself. Is there a possibility that the observed effect could be influenced by other factors affected by the corrector? When the corrector was applied to the cells, were any expected or unexpected differential gene expression changes observed?

While we do not directly measure whether Ipsen-17 has effects on other signaling processes, previous work has shown that Ipsen-17 treatment does not indirectly alter signaling kinetics such as receptor internalization (Wang et al., 2014). Furthermore, our analysis methods inherently account for this by normalizing variant effects to WT signaling levels. Any observed rescue of a given variant inherently means that the variant is specifically more responsive to Ipsen-17 than WT, and the fact that different variants exhibit different levels of rescue is reassuring that the mechanism is on target to MC4R. Lastly, Ipsen-17 is known to be an antagonist of alpha-MSH activity and is thought to bind directly to the same site on MC4R (Wang et al., 2014).

We have revised text in the Methods section as follows (see Running DMS Assays) to better articulate this : “For chaperone experiments, cells were washed 3x with 10 mL DMEM to remove Ipsen 17 prior to agonist stimulation as it has been shown to be an antagonist of α-MSH activity and is thought to bind directly to the same site on MC4R (Wang et al. 2014).”

(7) As mentioned in the introduction, gain-of-function (GoF) variants are known to be protective against obesity. It would be interesting to see further studies on the observed GoF variants. Do the authors have any plans for additional research on these variants?

We agree this would be an excellent line of inquiry, but due to changes in company priorities we unfortunately do not have any plans for additional research on these variants.

**Reviewer 2 (Public reviews):**
OverviewIn this manuscript, the authors use deep mutational scanning to assess the effect of ~6,600 protein-coding variants in MC4R, a G protein-coupled receptor associated with obesity. Reasoning that current deep mutational scanning approaches are insufficiently precise for some drug development applications, they focus on articulating new, more precise approaches. These approaches, which include a new statistical model and innovative reporter assay, enable them to probe molecular phenotypes directly relevant to the development of drugs that target this receptor with high precision and statistical rigor.They use the resulting data for a variety of purposes, including probing the relationship between MC4R's sequence and structure, analyzing the effect of clinically important variants, identifying variants that disrupt downstream MC4R signaling via one but not both pathways, identifying loss of function variants are amenable to a corrector drug and exploring how deep mutational scanning data could guide small molecule drug optimization.StrengthsThe analysis and statistical framework developed by the authors represent a significant advance. In particular, the study makes use of barcode-level internally replicated measurements to more accurately estimate measurement noise.The framework allows variant effects to be compared across experimental conditions, a task that is currently hard to do with rigor. Thus, this framework will be applicable to a large number of existing and future deep mutational scanning experiments.The authors refine their existing barcode transcription-based assay for GPCR signaling, and develop a clever "relay" new reporter system to boost signaling in a particular pathway. They show that these reporters can be used to measure both gain of function and loss of function effects, which many deep mutational scanning approaches cannot do.The use of systematic approaches to integrate and then interrogate high-dimensional deep mutational scanning data is a big strength. For example, the authors applied PCA to the variant effect results from reporters for two different MC4R signaling pathways and were able to discover variants that biased signaling through one or the other pathway. This approach paves the way for analyses of higher dimensional deep mutational scans.The authors use the deep mutational scanning data they collect to map how different variants impact small molecule agonists activate MC4R signaling. This is an exciting idea, because developing small-molecule protein-targeting therapeutics is difficult, and this manuscript suggests a new way to map small-molecule-protein interactions.WeaknessesThe authors derive insights into the relationship between MC4R signaling through different pathways and its structure. While these make sense based on what is already known, the manuscript would be stronger if some of these insights were validated using methods other than deep mutational scanning.Likewise, the authors use their data to identify positions where variants disrupt MC4R activation by one small molecule agonist but not another. They hypothesize these effects point to positions that are more or less important for the binding of different small molecule agonists. The manuscript would be stronger if some of these insights were explored further.ImpactIn this manuscript, the authors present new methods, including a statistical framework for analyzing deep mutational scanning data that will have a broad impact. They also generate MC4R variant effect data that is of interest to the GPCR community.Recommendations for the authors:(1) Page 7 - the Gq reporter relay system is clever. Could the authors include the original data showing that the simpler design didn't work at all, or at least revise the text to say more precisely what "not suitable due to weak SNR" means?

We added a panel (D) to Supplementary Figure 2 showing that the native NFAT reporter was ~10x weaker than the CRE reporter, and the relay system amplified the NFAT signal to be comparable to the CRE reporter:

(2) Page 7 - Even though the relay system gives some signal, it's clearly less sensitive/higher background than Gs. How does that play out in the quantitative analysis?—AND—(4) Page 10 - The Gq library had fewer barcodes per variant, and, as noted above, the Gq reporter doesn't work quite as well as the Gs one. It would be nice if the authors could comment on how these aspects of the Gq experiments affected data quality/power to detect effects.

Due to the reviewer's excellent suggestion, we updated Supplementary Figure 2B to better contextualize the quantitative effects of the difference in signal to noise ratio of the Gq versus the Gs reporter system (see changes below). These distributions show the Z-statistic for testing either each stop mutation (red) or all possible coding variants against WT. Thus, a |Z| > 1.96 corresponds to a p = 0.05 in a two-sided Wald Test. We can see that in the Gs reporter, 95% of the stops are nominally significantly different from WT (visualized above with the majority of the red distribution being < -1.96). Alternatively, only 64% of stops are nominally significantly different from WT in Gq. This implies that it will be more difficult to detect effects in the Gq system, especially those less severe than stops.

In addition to the overall signal to noise ratio being less in the Gq system, there were also less barcodes per variant (28 vs 56 barcodes per variant on average for Gq vs Gs). As demonstrated in Supplementary Figure 2C, the error bars on our estimates are related to the number of barcodes per variant (Standard Error ~ 1 / sqrt(Number of Barcodes), as shown in the plot below). This suggests that our estimates of mutant effects will be less certain in the Gq library than the Gs library. For example, the average standard error in the Gq library was 0.260 which was ~1.58 times larger than the Gs library's 0.165. Finally, we believe this further reiterates the power of our statistical framework, as it naturally enables formalized hypothesis testing that takes these errors into account when making comparisons both within reporters and across reporters.

(3) Page 9 - it would be nice to see the analysis framework applied to a few existing datasets from other types of assays, to really judge its performance. That's not the main point of this paper, and it's fine, but it would be lovely!

We agree with the reviewer and hope others apply our framework to their problems to further refine its utility and applicability! To that end, we’ve open-sourced it under a permissive license to help encourage the community to use it. Part of the challenge in applying it to other existing datasets is that few DMS experiments leverage variant-level replication through barcodes. While we re-analyzed an older DMS data from Jones *et al*. 2020 to produce the distributions in Supplementary Figure 2b, a more thorough comparison is outside the scope of this paper. That said, we have two additional manuscripts in preparation that leverage this framework to analyze DMS data in different proteins and assay types.

(5) Page 10 - In discussing the relationship of the data to ClinVar and AM, the authors use qualitative comparisons like "majority" and "typically." Just giving numbers would better help the reader appreciate how the data compare.

We added specific proportions for these statements to the text for the ClinVar and AlphaMissense comparisons as follows:

(See Results > Comprehensive Deep Mutational Scanning of MC4R): “For example, the majority (63.3%, 31/49) of human MC4R variants classified as pathogenic or likely pathogenic in ClinVar (Landrum et al., 2014) lead to a significant reduction of Gs signaling under low α-MSH stimulation conditions (significance threshold: false discovery rate (FDR) < 1%; Fig. 2C). Variants that are significantly loss-of-function in this condition are rarer in the human population, and more common human variants have no significant effect on MC4R function (significance threshold: FDR < 1%; Fig. 2D). Loss-of-function variants by our DMS assay are also typically (e.g., AlphaMissense: 93.4%, 1894/2028) predicted to be deleterious by commonly used variant effect predictors like AlphaMissense (Cheng et al., 2023) and popEVE (Orenbuch et al., 2023) (Supplementary Fig. 5).”

(6) Pages 10-12, Figures 2C, E. The data look really nice, but the correlation with clinvar and the Huang data is not perfect (e.g. many pathogenic variants are classified as WT and partial LoF variants too). Can the authors comment on this discrepancy? For ClinVar, they should say when ClinVar was accessed and also how they filtered variants. I would recommend using variants with at least 1 star. Provided they did use high-quality clinical classifications, do they think the classifications are wrong, or their data? The same goes for Huang.—AND—(7) Page 13 - similar to previous comments, I'm curious about the 5 path/likely path ClinVar variants that are not LoF in the assay. Are they high noise/fewer barcodes? Or does the assay just miss some aspect of human biology?

ClinVar data was accessed on January 5, 2024 (see Methods: Comparison to human genetics data and variant effect predictors). No annotation quality filtering was performed, and we have revised the text as follows to clarify this:

(see Methods > Comparison to Human Genetics Data and Variant Effect Predictors): “Pathogenicity classifications of MC4R missense and nonsense variants were obtained from ClinVar (Landrum et al., 2014) on January 5, 2024, and all available annotations were included in the analysis regardless of ClinVar review status metric.”

A substantial proportion of the discrepancy between our data and ClinVar is, as the reviewer suggests, likely due to low quality ClinVar annotations. Of the five variants that the reviewer notes were reported as pathogenic/likely pathogenic but did not result in loss of protein function in any of our DMS assays, two (V50M and V166I) have been reclassified in ClinVar to uncertain or conflicting interpretation since we accessed annotations in early 2024. An additional two of the five discrepant variants (Q43K and S58C) currently have 0 star ratings to support their pathogenic/likely pathogenic annotation. The remaining discrepant variant (S94N) has a 1 star rating supporting an annotation of “likely pathogenic.

The Huang et al. paper did an admirably thorough job of aggregating variant annotations from more than a dozen primary literature sources that each reported functional validation data for small panels of variants. However, one inherent limitation of this approach is that the resulting annotation classes are based on experiments that were carried out using inconsistent methods and/or scoring criteria. For example, classifications in the Huang et al. paper are based on an inconsistent mix of functional assay types (e.g., Gs signaling, Gq signaling, protein cell surface expression, etc.), and different variants were tested in different cell types (e.g., HEK293T, CHO, Cos-7, etc.). In principle, DMS assays should provide a more accurate assessment of the relative quantitative differences between alleles since each variant was tested using identical experimental conditions and analysis parameters.

That being said, while very good, our assays are likely missing or only indirectly reporting on at least some aspects of MC4R biology. For example, in addition to Gs and Gq signaling, MC4R interfaces with β-arrestin. Variants that are protective against obesity-related phenotypes have been shown to increase recruitment of β-arrestin to MC4R, and we did not directly assess this function.

(8) Page 15, Fig 3C - The three variants they highlight all have paradoxical changes in bias as a-MSH dose is increased (e.g. the bias inverts). I'm not a GPCR expert, but this seems interesting and a little weird. Perhaps the authors could comment on it?

We agree this is an interesting observation that deserves further study, but unfortunately is outside the scope of our priorities at the moment. As noted, all three highlighted variants in this region have a biased basal activity, and this bias inverts upon stimulation. While we don’t have a good explanation for why this would be the case, this phenomenon has been previously observed for 158R (Paisdzior et al., 2020). Our DMS data emphasizes how diverse biased effects can be and further highlights the importance of characterizing these effects. It would be interesting if further studies could elucidate the mechanistic basis for this behavior and how it may be related to G protein coupling in this region.

(9) Page 16 - I'm not familiar with the A21x1 formalism. For the general reader, maybe the authors could introduce this formalism.

Given the shared structural topology of GPCRs, others have developed a variety of numbering schemes to refer to where various variants are to allow more direct comparisons between different GPCRs. We use the GPCRDB.org numbering scheme (e.g., F202^5x4^) as it takes experimentally determined structures into account. Roughly speaking, the number preceding the “x” corresponds to which transmembrane domain (one through seven) or region the residue is located in. The numbers following the “x” correspond to where that residue is located in that region relative to a structurally conserved residue that is always assigned 50. For example F202^5x48^ means that F202 is located in the 5th transmembrane helix and is 2 residues before the most conserved M204^5x50^. We updated the text to clarify this accordingly:

(see Results > Structural Insights into Biased Signaling): “Upon ligand binding, W258 (W258^6x48^ in https://gpcrdb.org/ nomenclature, where 6 corresponds to the 6th transmembrane helix and 48 denotes 258 is 2 residues before the most conserved residue in that helix (Isberg et al., 2015)) of the conserved CWxP motif undergoes a conformational rearrangement that is translated to L133^3x36^ and I137^3x40^, of the conserved PIF motif (MIF in melanocortin receptors).”

(10) Page 17, Figure 3A - Since 137, 254, and 140 are not picked out on the structure, I have no idea where they are. If the authors want to show readers these residues, perhaps they could be annotated or a panel added. Since ~1 entire page of the manuscript is dedicated to this cascade, it might make sense to add a panel. Just amplifying the comment above as regards position 79, others were discussed in that paragraph but not highlighted.

We updated Supplementary Fig. 6C,D to label all of the listed residues on the protein structure for easy reference.